# A module to convert spectral to narrowband snow albedo for use in climate models: SNOWBAL v1.2

Christiaan T. van Dalum[1], Willem Jan van de Berg[1], Quentin Libois[2], Ghislain Picard[3], and Michiel R. van den Broeke[1]

[1]Institute for Marine and Atmospheric Research, Utrecht University, Utrecht, The Netherlands
[2]CNRM, Université de Toulouse, Météo-France, CNRS, Toulouse, France
[3]University Grenoble Alpes, LGGE (UMR5183), Grenoble, France

**Correspondence:** Christiaan van Dalum (c.t.vandalum@uu.nl)

**Abstract.** Snow albedo schemes in regional climate models often lack a sophisticated radiation penetration scheme and generally compute only a broadband albedo. Here, we present the Spectral-to-NarrOWBand ALbedo module (SNOWBAL, version 1.2) to couple effectively a spectral albedo model with a narrowband radiation scheme. Specifically, the Two-streAm Radiative TransfEr in Snow model (TARTES) is coupled with the European Center for Medium-Range Weather Forecasts (ECMWF) Integrated Forecast System (IFS), cycle 33R1, atmospheric radiation scheme based on the rapid radiation transfer model, which is embedded in the regional climate model RACMO2. This coupling allows to explicitly account for the effect of clouds, water vapour, snow impurities and snow metamorphism on albedo. Firstly, we present a narrowband albedo method to project the spectral albedos of TARTES onto the 14 spectral bands of the IFS shortwave radiation scheme using a representative wavelength (RW) for each band. Using TARTES and spectral downwelling surface irradiance derived with the DIScrete Ordinate Radiative Transfer atmospheric model, we show that RWs primarily depend on the solar zenith angle (SZA), cloud content and water vapour. Secondly, we compare the TARTES narrowband albedo, using offline RACMO2 results for South Greenland, with the broadband albedo parameterizations of Gardner and Sharp (2010), currently implemented in RACMO2, and the multi-layered parameterization of Kuipers Munneke et al. (2011, PKM). The actual absence of radiation penetration in RACMO2 leads on average to a higher albedo compared with TARTES narrowband albedo. Furthermore, large differences between the TARTES narrowband albedo and PKM and RACMO2 are observed for high SZA and clear-sky conditions, and after melt events when the snowpack is very inhomogeneous. This highlights the importance of accounting for spectral albedo and radiation penetration to simulate the energy budget of the Greenland ice sheet.

## 1 Introduction

The absorption of shortwave solar radiation is a major contributor to the energy budget of snow and ice (Van den Broeke et al., 2005; Gardner and Sharp, 2010; Stroeve et al., 2013; He et al., 2018). Over fresh snow this energy absorption is limited due to the high reflectivity, i.e. albedo, of fine grained snow, but it becomes significant for darker older snow and glacial ice. This strong dependency of the albedo to the surface snow and ice properties is the driving mechanism of the melt-albedo feedback (e.g. Van As et al. (2013)). A warm event leads to rapid metamorphism of snow, which leads to coarser and less reflective snow

grains, which increases the energy available for heating and enhances snow metamorphism even more. Accurate representation of the albedo is thus essential for a reliable estimate of the energy budget, grain size and melt-albedo feedback of a snowpack (Picard et al., 2012; Van Angelen et al., 2012).

The penetration through and absorption of solar energy in snow and ice depends on the solar zenith angle, atmospheric conditions, and physical properties of snow, for example, grain size, grain shape and impurity concentration (Stroeve et al., 1997; Klok et al., 2003; Warren and Brandt, 2008; Gardner and Sharp, 2010; Libois et al., 2013, 2014; Dumont et al., 2014; Dang et al., 2015; Tedesco et al., 2016). Recent studies (Picard et al., 2009; Libois et al., 2013) show that grain shape also has a high impact on radiation penetration through snow. Besides, the optical properties of snow and ice are also highly wavelength dependent. The spectral albedo, i.e. the surface reflectivity for a wavelength, is highest for near-ultraviolet (near-UV, 300-400 nm), visible and near-infrared (near-IR, 750-1400 nm) radiation, but low and fluctuating in the IR part of the electromagnetic spectrum (Warren and Wiscombe, 1980; Ackermann et al., 2006; Warren et al., 2006; Gardner and Sharp, 2010; Dang et al., 2015; Picard et al., 2016). Furthermore, blue and near-UV radiation can penetrate a few meters into snow, while near-IR radiation does not reach deeper than a few millimeters (Picard et al., 2016). Also, a distinction has to be made for the albedo of direct radiation, which varies as a function of the solar zenith angle (SZA), and of diffuse radiation. Although broadband albedo, defined as the ratio of upwards to downwards shortwave radiative flux on a horizontal surface integrated over the solar spectrum, or two-band albedo parameterizations often neglect internal heating due to radiation penetration, they are commonly used in global and regional atmospheric models (Aoki et al., 2011), but are prone to inaccuracies due to these simplifications. Moreover, cloud cover and water vapour have opposite impacts on the incoming spectrum than a high SZA, i.e., the spectrum shifts towards visible light for cloud cover and water vapour, and towards the IR for a high SZA (Dang et al., 2015). Impurities mostly affect the reflectivity for near-UV and visible light, while snow metamorphism mostly affect the reflectivity for near-IR light (Tedesco et al., 2016). The grain radius of impurities determines the scattering regime. The typical grain radius of soot and humic-like substances (HULIS) are small compared to shortwave wavelengths, while the typical grain radius of dust is not small. Consequently, an albedo model has to be compatible for Rayleigh scattering to incorporate soot and HULIS, and for Mie theory for dust and biological material (Tegen and Lacis, 1996; Cook et al., 2017). It is clear that an interactive spectral albedo and radiation penetration model are required to derive correct estimates for the solar energy absorption in snow and ice and its impact on the snow heating rates (Flanner and Zender, 2006).

In this paper, we discuss implementation strategies of a spectral albedo and radiation penetration scheme in the polar version of the regional climate model RACMO2.3p2 (Noël et al., 2018), from now on called RACMO2, and compare results with the broadband albedo parameterization of Gardner and Sharp (2010, GS) and the multi-layered broadband albedo parameterization of Kuipers Munneke et al. (2011, PKM). Currently, GS, extended with an elevation correction as described by PKM, is implemented in RACMO2, hence RACMO2 does neither include a sophisticated radiation penetration scheme, nor spectral albedo. Therefore, we discuss the coupling of the radiative transfer model Two-streAm Radiative TransfEr in Snow (TARTES, Libois et al., 2013) with the ECMWF radiation scheme McRad based on the shortwave Rapid Radiation Transfer Model (RRTM$_{sw}$, Mlawer et al., 1997; Clough et al., 2005; Morcrette et al., 2008; ECMWF, 2009), which is embedded in RACMO2. RRTM$_{sw}$ calculates solar radiation fluxes using fourteen contiguous shortwave bands. TARTES is a model based on the asymptotic

analytical radiative transfer theory (Kokhanovsky, 2004) and the two-stream approximation of the radiative transfer equation (Jiménez-Aquino and Varela, 2005), and allows for a multi-layer, heterogeneous snow profile. TARTES returns a spectral albedo and a radiation absorption profile within the snowpack.

Coupling $RRTM_{sw}$ with TARTES is not trivial. It is computationally too expensive to run TARTES for the hundreds to thousands wavelengths that are required to derive the spectral albedo of a snow profile in full detail. Moreover, the spectral irradiances are lacking in RACMO2 to convert the spectral albedos to a narrowband albedo, i.e. the weighted albedo of a spectral band. Neither can TARTES be run in a broadband mode equivalent to $RRTM_{sw}$. Therefore, fourteen predefined representative wavelengths are determined, which leads to correct albedos for each spectral band in $RRTM_{sw}$. A representative wavelength (RW) depends on the irradiance distribution and albedos within a spectral band, which in turn depends on atmospheric conditions, e.g., liquid water clouds, ice clouds, water vapour, aerosols, and the SZA (Leckner, 1978; Gueymard, 2001; Hussain, 1984). In order to determine the RWs, TARTES is coupled with the libRadtran software package (Mayer and Kylling, 2005) using the Discrete Ordinates Radiative Transfer Program for a Multi-Layered Plane-Parallel Medium (DISORT, Stamnes et al., 1988, 2000) and this combination has been run for different atmospheric and solar conditions in high spectral detail. The coupling framework forms the new Spectral-to-NarrOWBand ALbedo module (SNOWBAL, version 1.2).

The remainder of this paper consists of six sections, starting with a description of the method and models used in section two. Section three discusses the optimization method of predefining RWs to assess the relevant quantities to consider. Section four deals with the numerical particularities required for the implementation. Section five presents the first results of an offline coupling case of TARTES using RACMO2 output for South Greenland in 2007, and compares this with results from the broadband albedo schemes of RACMO2 and PKM. The final sections discuss our findings, summarize the results and provide an outlook.

## 2    Method

In this section, the models RACMO2, TARTES and DISORT are described, and the method to couple TARTES with RACMO2 and its numerical implementation is discussed.

### 2.1    Regional climate model

The polar version of the Regional Atmospheric Climate Model (RACMO2), version 2.3p2, combines the physics for surface and atmospheric processes from the European Center for Medium-Range Weather Forecasts (ECMWF) Integrated Forecast System (IFS), cycle 33R1 (ECMWF, 2009), with the atmospheric dynamics from the High Resolution Limited Area Model (HIRLAM), version 5.0.3 (Unden et al., 2002). RACMO2 is developed at the Royal Netherlands Meteorological Institute (KNMI) and the polar version is further adapted for glaciated regions at the Institute for Marine and Atmospheric Research (IMAU), by including a dedicated glaciated tile that describes in detail the ice-atmosphere interaction and the snow and ice processes in the snow column (Noël et al., 2015).

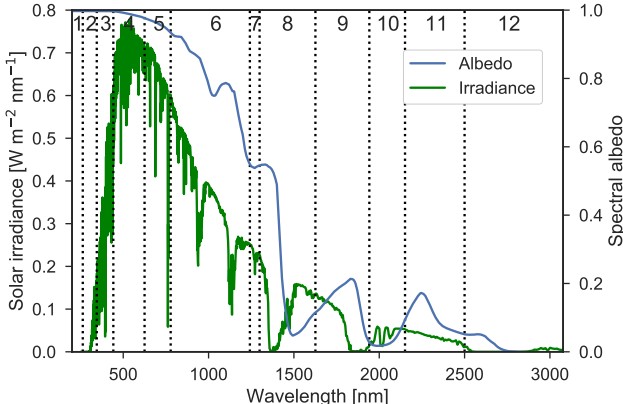

**Figure 1.** Direct shortwave irradiance and albedo as a function of wavelength for a fresh snow layer, clear-sky conditions and a SZA of $53°$. The albedo is derived with TARTES and the irradiance by DISORT. The first twelve shortwave spectral bands of RACMO2 are numbered and indicated by vertical dashed lines. Band 13 and 14 are left out as they have a negligibly small irradiance and a snow albedo close to 0.

Shortwave radiation in RACMO2 is computed by the embedded atmospheric radiation scheme of the ECMWF, RRTM$_{sw}$,
which uses fourteen contiguous spectral bands, ranging from 200 to 12500 nm (Figure 1). These bands are constructed around the important absorption lines for the atmosphere (Morcrette et al., 2008). The sub-band albedo and energy distribution can vary considerably, e.g. band 6 and 8 are wide and contain significant fractions of the incoming radiation (Figure 1). RRTM$_{sw}$ computes instantaneous flux profiles for clear-sky and total-sky conditions.

The current snow albedo scheme for glaciated gridboxes is a plane-parallel, two-layer broadband albedo parameterization,
which is based on the parameterization of Gardner and Sharp (2010, GS) and includes the effect of grain radius, solar zenith angle, cloud cover and impurities on albedo. Kuipers Munneke et al. (2011) further enhanced the parameterization by adding an altitude-dependent term for the optical thickness during clear-sky conditions. The GS albedo scheme indirectly depends on wavelength using tuning parameters. Albedo computations are limited to the first two snow or ice layers, with the lowest layer considered as a semi-infinite background layer. In theory, this incorporates the effect of radiation penetration. However,
in practice radiation penetration is neglected as the uppermost two layers are together typically less than 8 cm thick, but for fresh snow conditions often much less, and only the uppermost layer defines the albedo in case of fine grained snow with some impurities (Gardner and Sharp, 2010, Eq. 13). The albedo scheme of Kuipers Munneke et al. (2011, PKM) is similar to GS, with the sole difference that PKM uses multiple snow layers and derives the albedo from the model layer albedos using an exponentially decreasing factor. However, PKM is not embedded in RACMO2 and is used for comparison only. In RACMO2,
the effective grain radius $r_{eff}$ defines the grain size, implying spherical particles. The effective grain radius relates to the specific surface area (SSA) with

$$\text{SSA} = \frac{3}{r_{\text{eff}} \rho_{\text{ice}}}, \tag{1}$$

with $\rho_{\text{ice}}$ the density of ice (917 kg m$^{-3}$). Snow metamorphism, i.e. the growth of $r_{\text{eff}}$ in time, is modeled using the dry and wet snow growth rates (Brun et al., 1989; Flanner and Zender, 2006).

RACMO2 lacks impurity parameterizations for snow and ice that can model the large spatial and temporal variability of e.g. soot in snow and cryoconite in glacial ice (Bory et al., 2002; Tedesco et al., 2016). In RACMO2, the soot concentration in snow is temporally and spatially constant in a simulation.

## 2.2   Radiative transfer model

The Two-streAm Radiative TransfEr in Snow model (TARTES, Libois et al., 2013) provides the spectral albedo and irradiance
profile of a multi-layered snowpack for any given wavelength between 199 and 3003 nm (e.g. Figure 1). Radiative fluxes are calculated for both direct and diffuse radiation, using the $\delta$-Eddington approximation (Joseph et al., 1976). Diffuse radiation is approximated as a direct beam at a SZA of 53 degrees. The refractive index of ice, which is calculated by Warren and Brandt (2008), but also physical properties of a snowpack (e.g. density, grain radius, grain shape, impurity load) and the SZA for direct radiation, have to be prescribed. The grain size is expressed in terms of SSA, and grain shape is determined by a
geometrical optics asymmetry parameter $g^G$, with the total asymmetry parameter $g = \frac{1}{2}(1 + g^G)$ for non-absorbing particles, and an absorption enhancement parameter $B$. This absorption enhancement parameter represents the changing photon path length within a grain due to internal reflections. These parameters allow for complex grain shapes. The parameters $B$ and $g$ can be computed for spheres and prove to be quite successful for albedo calculations (Gallet et al., 2009; Grenfell and Warren, 1999), but much less for transmittance or penetration depth simulations. Libois et al. (2014) demonstrate that $g$ cannot be
determined based on optical measurements, because it is coupled to SSA, and must be assumed somehow. The relative success of spheres for albedo calculations, which depends on $B/(1 - g^G)$, means that any shape such that $B/(1 - g^G)$ equals that of spheres should be quite efficient for albedo simulations. Hence, the best estimate of $g$ would be such that $B/(1 - g^G)$ equals the value for spheres. As TARTES is a spectral model, it does not allow for narrowband or broadband calculations.

Unless stated otherwise, we evaluate TARTES on a four-layered snowpack, with layer thicknesses of: 0.2, 0.5, 1.0, and 3.0
m from top to bottom, respectively. The density and SSA per layer are, from top to bottom, 200, 300, 350 and 450 kg m$^{-3}$ and 40, 15, 10 and 3.0 m$^2$ kg$^{-1}$, respectively. These layers represent a winter fresh snowpack on top of melted snow of the previous year. As discussed later in detail, the sensitivity of the resulting snowpack to the exact snowpack properties results is low.

## 2.3   DISORT

The Discrete Ordinate Radiative Transfer (DISORT) solver (Stamnes et al., 1988, 2000) computes a net shortwave radiation flux at the surface for direct and diffuse radiation for atmospheric conditions that are prescribed using the libRadtran software package (Mayer and Kylling, 2005). DISORT considers a plane parallel, scattering and absorbing atmosphere for a monochromatic and unpolarized beam of light at a given angle. The pseudo-spherical variant SDISORT (Dahlback and Stamnes, 1991) also accounts for atmospheric curvature, which is particularly relevant for high SZA. SDISORT is used in this paper and is
called DISORT from now on unless stated otherwise. Thirty-two streams, i.e. computational polar angles, are used to solve

the radiative transfer equation (Stamnes et al., 2000). Net shortwave radiative fluxes are computed for wavelengths between 250 and 3077 nm in 2450 spectral bands. These bands have a high resolution for visible light, i.e. in the order of a few tenths of a nanometer, and a lower resolution for near-IR radiation, i.e. in the order of a few nanometers. For the runs presented here, a subarctic winter atmospheric profile is chosen, which is one of the Air Force Geophysics Laboratory (AFGL) standards (Anderson et al., 1986), but the water vapour load is changed according to conditions. Rural type aerosols during winter are set up by the aerosol model by Shettle (1990) for a 2 km thick boundary layer and a background aerosol load above the boundary layer, but the sensitivity of the aerosol load to the results is low. Both the subarctic winter atmospheric profile and the aerosol type and load are the closest representation for the Arctic that we can prescribe in the libRadtran package. The surface albedo of DISORT is prescribed in libRadtran with a spectral albedo profile of TARTES. The surface albedo does not, per definition, matter for the direct downward radiative flux. As diffuse radiation is approximated as a direct beam with a SZA of 53 degrees by TARTES, such an albedo curve should be suitable for all SZAs for the diffuse downward radiative flux. Still, some part of the diffuse flux is due to direct radiation that is scattered back by the atmosphere after reflecting at the surface first. For this part, a spectral albedo curve depending on SZA would be more suitable, but would have a second order effect on DISORT with respect to other variables. Therefore, a spectral albedo profile of TARTES using a SZA of 53 degrees for the reference snowpack is sufficient enough to use as the surface albedo of DISORT.

A liquid water or ice content in the atmosphere can be inserted manually at a chosen height. The effective droplet radius for ice and water clouds are set to 20.0 µm and 10.0 µm, respectively, which are realistic radii for clouds in the Arctic (Stubenrauch et al., 2013; Fitzpatrick et al., 2004; Mahesh et al., 2001; Fu, 1996; Key et al., 2002; King et al., 2004). The entire cloud ice content, i.e. the vertically integrated ice water content, or ice water path (IWP), is homogeneously located in a layer between 6 and 7 km above the surface, which is a typical height for high Arctic ice clouds (Garrett et al., 2001). The cloud liquid water concentration is homogeneously located in a layer between 2.0 to 4.0 km above the surface. For vertically integrated liquid water content, or liquid water path (LWP) above 10.0 kg m$^{-2}$, the liquid water concentration is distributed between between 2.0 and 6.0 km to prevent excessively high concentrations. SDISORT does not provide reliable fluxes for clouds with LWP or IWP > 0.5 kg m$^{-2}$, hence the regular DISORT solver is used instead for these cases.

## 2.4 SNOWBAL

There is no simple method to couple TARTES with RACMO2. If the spectral albedo, which is from now on simply called the albedo, is determined in detail using many wavelengths (e.g. Figure 1), this would lead to a significant numerical burden, although it is likely possible to parameterize this spectral curve using in the order of thirty well-chosen spectral albedos. Moreover, it would not deliver accurate narrowband albedos as the impact of sub-band irradiance variations is not taken into account. Similarly, using the median wavelength would be computationally more efficient, but inaccurate. Therefore, we determine representative wavelengths (RWs) depending on atmospheric conditions (e.g. LWP, IWP) and SZA for each band. Some bands are broad, thus splitting these spectral bands up in smaller intervals has been considered to improve the representation of sub-band albedo variability. However, RACMO2 does not compute sub-band energy fluxes. Hence, even with an efficiently derived fully spectral snow albedo or with smaller spectral bands, it would not be possible to estimate the

snow albedo accurately within RACMO2 as sub-band energy fluxes are essential, but unavailable. Therefore, we discard this approach.

For the computation of RWs, a weighted mean of the sub-band albedo is computed, using the solar energy distribution within a band provided by DISORT. This weighted mean of a band then corresponds with a certain representative wavelength. For this wavelength, TARTES would produce exactly the weighted mean albedo. This coupling framework forms the Spectral-to-

NarrOWBand ALbedo module (SNOWBAL, version 1.2).

In many bands, e.g. band 6 or 8, the albedo is not a continuously decreasing function (Figure 1). Two or more wavelengths can therefore represent the same weighted mean albedo. To avoid interpolation errors that would lead to erroneous narrowband albedos, we made sure that the RWs remained in a spectral interval over which the albedo is continuously increasing or decreasing.

For the determination of the RW, we evaluate the relevance of the snow profile, SZA, aerosol concentration, direct to diffuse ratio and the liquid and ice water content in the atmosphere. We select the most relevant parameters to make a scheme of RWs. Computational time and memory usage in RACMO2 limits the number of variables that can be used to compute the RW to preferably no more than three.

The coupling of TARTES with RACMO2 requires a lookup table of RWs as a function of the relevant variables for each

of the first twelve spectral bands. The snow and ice albedo of the last two bands, ranging from 3077 - 3846 nm and 3846 - 12500 nm, is very low. Therefore, the albedo is set to zero and no RW is required. The most suitable RW is determined by linear interpolation. These RWs are then given to TARTES, which computes albedos and a radiation absorption profile for each spectral band, which is then given back to RACMO2 for further calculations.

TARTES distinguishes a direct and diffuse part of the incoming flux, while RACMO2 computes clear-sky and total-sky

fluxes. Therefore the clear-sky flux has to be split into a diffuse and direct radiation part. For this, the direct to diffuse ratio of incoming solar radiation as calculated by DISORT, is determined as a function of the SZA. This direct to diffuse ratio is implemented in RACMO2 in the form of a lookup table. Using the modeled cloud fraction, the aforementioned total-sky flux can be split into a clear-sky part, which uses the procedure described above, and an overcast part. Usually, the direct radiation in the overcast part of the total-sky flux is negligible and all radiation can be considered as diffuse, so only for very thin clouds

is a direct radiation flux included.

## 2.5   Model simulation settings

In order to offline compare the narrowband albedo computed by TARTES with GS as implemented in RACMO2 and PKM, a one-year simulation, 2007, for South Greenland on a 20 km grid has been carried out. This simulation is driven at the boundary by ERA-Interim reanalysis data (Dee et al., 2011). As the implementation of impurities is fundamentally different in

TARTES compared to RACMO2 and PKM, clean snow is used in the simulation presented here, in order to prevent unnecessary complication of the model intercomparision. Furthermore, because the focus of this study is on the albedo of snow, the albedo of glacial ice is prescribed. Glacial ice, if clean, has an albedo of about 0.5, but due to various types of impurities the albedo of glacial ice in Greenland varies between 0.2 and 0.5, with a typical value of 0.4. In normal simulations with RACMO2 (Noël

et al., 2018), a spatially varying, but time constant glacial ice albedo derived from MODIS data is used. Here, a temporally and
spatially constant glacial ice albedo of 0.4 is used in RACMO2 and PKM. In TARTES, a similarly low albedo is achieved by
representing glacial ice by snow with a SSA of 0.08 $m^2$ $kg^{-1}$, i.e. a $r_{eff}$ of 0.04 m, which is relatively close to the SSA of 0.16
$m^2$ $kg^{-1}$ found by Dadic et al. (2013) for ice at 10 m depth. As TARTES still evaluates the effect of SZA and clouds on the
glacial albedo, this albedo is not constant in time as in RACMO2 and PKM. The asymptotic analytical radiative transfer theory
(Kokhanovsky, 2004) in TARTES is valid for snow and firn, but it is invalid for a strong absorbing medium, so the results over
bare ice are only indicative. Moreover, radiation scattering in pure ice only takes place at ice-bubble boundaries, and therefore
only depends on the bubble size (Gardner and Sharp, 2010), which is not taken into account in TARTES.

## 3 Representative wavelength

In this section, we assess the impact of various physical properties on the RW.

### 3.1 Solar zenith angle

The albedo of snow depends strongly on the solar zenith angle (SZA) (Shupe and Intrieri, 2004; Liu et al., 2009). The SZA
affects the albedo in two ways. Firstly, an increasing SZA leads to a longer path through the atmosphere for light to travel,
resulting in a higher chance to scatter and thus decreasing the incoming radiation and a relative increase of the diffuse flux
(Figure 2). In addition, the spectrum of the irradiance is shifted to larger wavelengths for higher SZA, because Rayleigh
scattering is more effective for smaller wavelengths. Secondly, for clear-sky conditions an increasing SZA results in a shallower
angle of incidence in the snowpack and less vertical penetration in the snow. As a result, light is scattered more easily out of
the snowpack and the impact of the upper snow layers becomes more important than the deeper layers. These upper layers are
often fresh snow layers, characterized by strong scattering, and thus the net albedo will increase (Gardner and Sharp, 2010).

Figure 2a shows the albedo computed with TARTES for direct radiation and the shortwave irradiance computed by DISORT
for three values of SZA. The figure indicates large sub-band variations in the IR part of the spectrum, e.g. band 6 and 8 (see
also Figure 1), leading to weighted mean albedo differences between SZAs of those bands, and therefore also for the RW.
Consequently, it is imperative to take sub-band deviations into account when considering SZA. The impact of SZA on the
albedo of diffuse radiation is less pronounced (Figure 2b). In TARTES, diffuse radiation is modeled as an incoming direct
solar beam at a SZA of 53 degrees. As a result, the path for light to travel through the snow is not changed. Still, the spectral
distribution of the incoming flux is altered, which affects the narrowband albedo slightly.

In order to show the added value of narrowband albedos and the performance gain by using RWs, the root-mean-square error
(RMSE) between estimated narrowband albedos derived with several approximation methods and the best estimate, derived
with fully spectral DISORT and TARTES calculations, are analyzed. The RSMEs are weighted using the band irradiance for
that specific situation in order to emphasize relevant narrowband albedo deviations. Small albedo differences may have large
unforeseen consequences on the energy budget and subsequent development of the snowpack, so only a RMSE of 0.01 or lower
is deemed acceptable. All the tests discussed below use the default atmospheric conditions and snow properties.

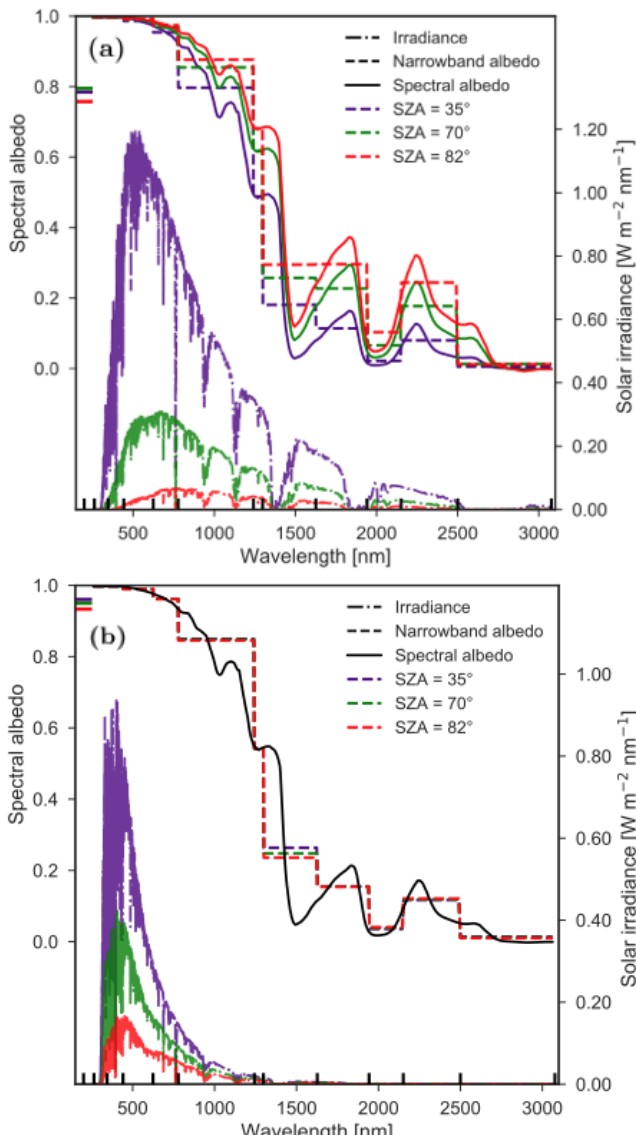

**Figure 2.** Irradiance and albedo as a function of wavelength for clear-sky conditions for (**a**) direct and (**b**) diffuse radiation for SZA = 35.33°, SZA = 70.67° and SZA = 82.44°. The albedo is derived by TARTES for a fresh snow layer and the irradiance by DISORT. The narrowband albedo for each of the first twelve spectral bands of RACMO2 is indicated by the dashed line. The black vertical lines on the x-axis indicate the spectral band edges of RACMO2. The horizontal coloured lines on the y-axis indicate the weighted broadband albedo.

Firstly, broadband albedos as a function of SZA are tested. The broadband albedo, which is for direct radiation and for the atmospheric and snow conditions described in the method section close to 0.78 for most SZAs (except for high SZA), and is used to compute a RMSE for each band. The RMSE is very high, i.e. larger than 0.1, for all spectral bands except band

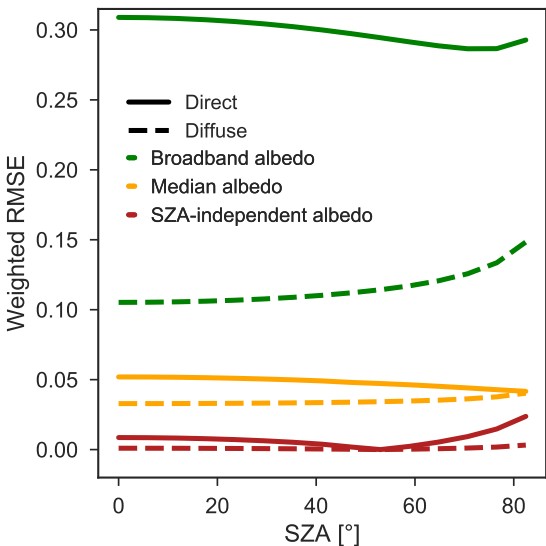

**Figure 3.** Weighted albedo root-mean-square error (RMSE) of the estimated albedos with respect to the narrowband albedos derived from the fully spectral TARTES-DISORT calculations, for direct and diffuse irradiance. The SZA-independent albedo is computed at SZA = 53°.

6, because the spectral albedo is much higher than the broadband albedo for bands 1 to 5 and much lower for bands 7 to 12

80 (Figure 1). Consequently, the weighted RMSE is very high. Therefore, neglecting the spectral dependency of albedo leads to significant errors, irrespective of the SZA (Figure 3). Lower RMSEs are found for diffuse than for direct irradiation, as the diffuse radiation is more concentrated at smaller wavelengths where the snow albedo is high and rather constant. Next, the RSME is analyzed if the narrowband albedo is represented by the albedo of the median wavelength of the band. Including the spectral variation of snow albedo in this very simplified manner already reduces the RMSE strongly compared to a broadband albedo description, but remains inaccurate, i.e. larger than 0.01. Clearly sub-band variations in irradiance and albedo have a

strong impact on the narrowband albedo. Thirdly, RWs are tested, but without a dependency of the RW to the SZA, as an insignificant dependency would allow a simplified implementation of TARTES in RACMO2. For this test, the RWs for a SZA of 53 degrees are used, as this is the SZA of diffuse radiation in TARTES. Consequently, the diffuse RWs are correct for this SZA and thus the RMSE is 0. For other SZAs, the RMSE is one order of magnitude less than if the median wavelength of the band is used, but the RMSE is still significant. Of course, the reference SZA can be optimized to provide correct narrowband

albedos for low or high SZA, but the results presented in Figure 2 show that in any case large errors would be made, if we move away from this reference SZA. For diffuse radiation, the RMSE is small (i.e. < 0.01) as it is only driven by variations in the distribution of incoming energy within bands. Concluding, for clear-sky conditions, RWs must be a function of the SZA for accurate narrowband albedo estimates.

## 3.2 Aerosols and preciptable water

For cloud free conditions, other physical properties than SZA potentially impact the RW. In the following sections, we assess whether these properties must be taken into account. Below, we compare the narrowband albedos obtained using the RWs as function of the SZA derived with the default TARTES and DISORT settings as a reference and compare these narrowband albedos with those for altered atmospheric conditions.

Aerosols significantly influence the incoming solar radiation at the surface (Satheesh and Moorthy, 2005). Here, rural winter
aerosols for a subarctic winter atmospheric profile are prescribed in the libRadtran package. Assuming an aerosol-free atmosphere leads to a higher RMSE, but is still acceptable, i.e. $< 0.01$. This indicates that changing the aerosol load does not lead to a relevant narrowband albedo change.

The effect of water vapour on the solar spectrum is considerable and has been known for a long time (Abbot, 1911), and as such, the impact of the vertically integrated water vapour (IWV) on the narrowband albedo has to be investigated. The IWV of
the default subarctic winter profile is relatively low and about $4 \, \mathrm{kg \, m^{-2}}$ (Anderson et al., 1986). Although this IWV is common in Greenland, it is spatially variable and can increase considerably during summer (Castellani et al., 2015). In libRadtran, the IWV can be varied while still retaining the relative vertical distribution of water vapour of the subarctic winter. The IWV is varied for both direct and diffuse radiation for clear-sky conditions (Figure 4). For direct radiation, the impact on mostly the near-IR irradiance is significant and consequently also on the narrowband and broadband albedo. The broadband albedo for
2, 10 and $40 \, \mathrm{kg \, m^{-2}}$ IWV is 0.785, 0.803 and 0.820 respectively. Diffuse radiation for both clear-sky and cloudy conditions, on the other hand, is affected only to a limited extent. Although the narrowband albedo is altered considerably for band 8, the impact on the broadband albedo is limited, because most of the near-IR radiation is already filtered out. The broadband albedo for clear-sky diffuse radiation for 2, 10 and $40 \, \mathrm{kg \, m^{-2}}$ IWV is 0.956, 0.951 and 0.956 respectively. In addition, the broadband albedo for subarctic winter with default IWV is 0.950. The variations in broadband albedo for clear-sky diffuse
radiation are considerably less than for direct radiation. Similar variations are also found for cloudy conditions. In conclusion, it is only necessary to take IWV into account for direct radiation, while IWV can be safely omitted for diffuse radiation for both clear-sky and cloudy conditions. Therefore, the default subarctic winter profile is used for diffuse radiation for clear-sky and cloudy conditions.

## 3.3 Cloud cover

Both ice and water clouds are known to strongly affect the incoming solar radiation in intensity, spectral distribution and angular distribution (Warren, 1982; Gardner and Sharp, 2010; Dang et al., 2015), and alter the broadband albedo, but have not been taken into account yet. Even a thin cloud nullifies the direct irradiance in favor of the diffuse irradiance. Figure 5 illustrates the effect of cloud cover on the diffuse spectral irradiance at the surface and the albedo. In this figure, liquid clouds are analyzed and the SZA of diffuse light is constant at 53 degrees. For clear-sky conditions (LWP $= 0.0 \, \mathrm{kg \, m^{-2}}$), a large part of
the incoming solar radiation at the surface is IR, for which the albedo is very low. With a gradual thickening of clouds, almost all IR radiation is filtered out and the broadband albedo increases. The shift towards shorter wavelengths results in a shift of

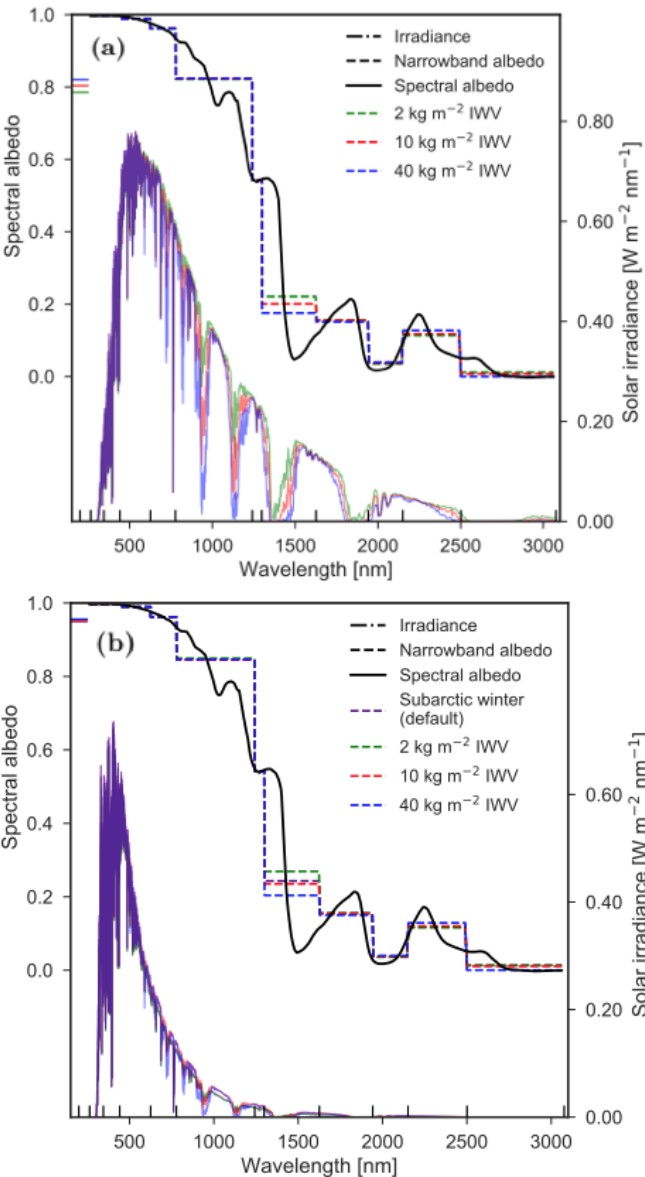

**Figure 4.** Irradiance and albedo as a function of wavelength for clear-sky conditions for various IWV values for (**a**) direct and (**b**) diffuse radiation for SZA = 53°. The albedo is derived by TARTES for a fresh snow layer and the irradiance by DISORT. The narrowband albedo for each of the first twelve spectral bands of RACMO2 is indicated by the dashed line. The black vertical lines on the x-axis indicate the spectral band edges of RACMO2. The horizontal coloured lines on the y-axis indicate the weighted broadband albedo. The water vapour is distributed vertically in the same manner as the default subarctic winter, which has a IWV of approximately 4 kg m$^{-2}$.

the RW as well, causing the narrowband albedo to strongly vary as function of LWP. For example, the narrowband albedo of

band 6, which ranges between 778 and 1242 nm and contains a large amount of energy, gradually increases from 0.85 to 0.93. Similar results are found for ice clouds. Concluding, such large differences in narrowband albedo cannot be neglected, so the

RW must be a function of LWP.

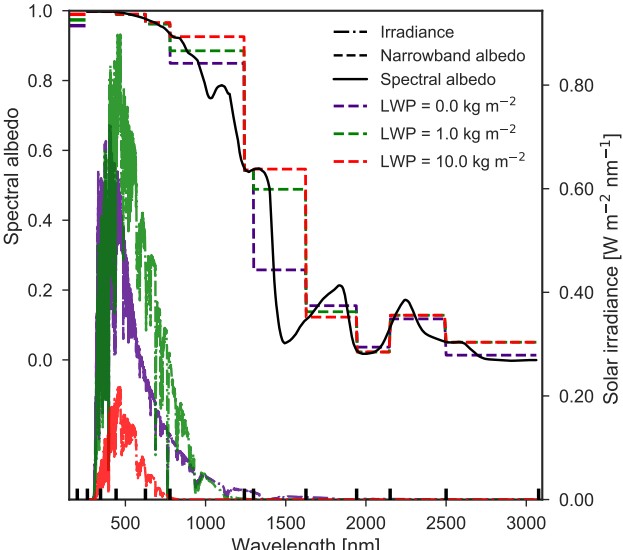

**Figure 5.** Diffuse spectral irradiance at the surface computed by DISORT, and spectral albedo, computed by TARTES, as a function of wavelength. The narrowband albedo for each of the first twelve spectral bands of RACMO2 is indicated by the dashed lines for LWP = 0.0, 1.0 and 10.0 kg m$^{-2}$. The black vertical lines on the x-axis indicate the edges of the spectral bands of RACMO2. The horizontal coloured lines on the y-axis indicate the mean broadband albedo, weighted over the entire spectrum.

To test whether SZA, LWP and IWP are all relevant factors for the RW for diffuse irradiation, we assess the errors made if those factors are neglected. For example, Figure 6a shows the weighted RMSE of the derived narrowband albedos as a function of IWP and SZA, with a constant LWP of 0.1 kg m$^{-2}$. Hence, it shows whether RWs derived when LWP = 0.1 kg m$^{-2}$ and IWP = 0.0 kg m$^{-2}$ would also provide realistic narrowband albedos for other values of IWP and SZA. Again, a RMSE of 0.01

or lower is deemed acceptable. In Figure 6a, a limited dependency of the RSME to SZA is found, only for very high SZA larger deviations arise. More important are the high RSMEs for IWPs between 0.1 and 1.0 kg m$^{-2}$, which are typical IWPs around the southeastern coastal zone of Greenland (Van Tricht et al., 2016). For low IWPs (IWP < 0.1 kg m$^{-2}$), the RWs derived for IWP = 0 kg m$^{-2}$ are still more-or-less valid, but for increasing IWP this assumption no longer holds. For these IWPs, near-IR light is partly removed from the incoming spectrum, so the narrowband albedos of bands 6 and 8 have risen substantially (similarly

as shown in Figure 5), while these bands still receive a significant fraction of the irradiance. For IWPs larger than 1 kg m$^{-2}$, irradiance is concentrated to wavelengths smaller than 800 nm for which the snow albedo is high. Hence, the RWs of spectral

bands with wavelengths above 800 nm might be inaccurate, but they are of little relevance for the computation of a broadband albedo.

Figure 6b shows a case where the effect of SZA on RW is left out, but the RWs do depend on the IWP. For the RWs, a SZA of 53 degrees is used. As a result, the weighted RMSE is zero for this SZA. Similar to Figure 6a, LWP = 0.1 kg m$^{-2}$ and IWP and SZA are varied. For low and high SZA, the weighted RMSE is nonzero, but negligible as maximum values are 0.002. Hence, the SZA has a limited impact on the RW and narrowband albedos for overcast conditions.

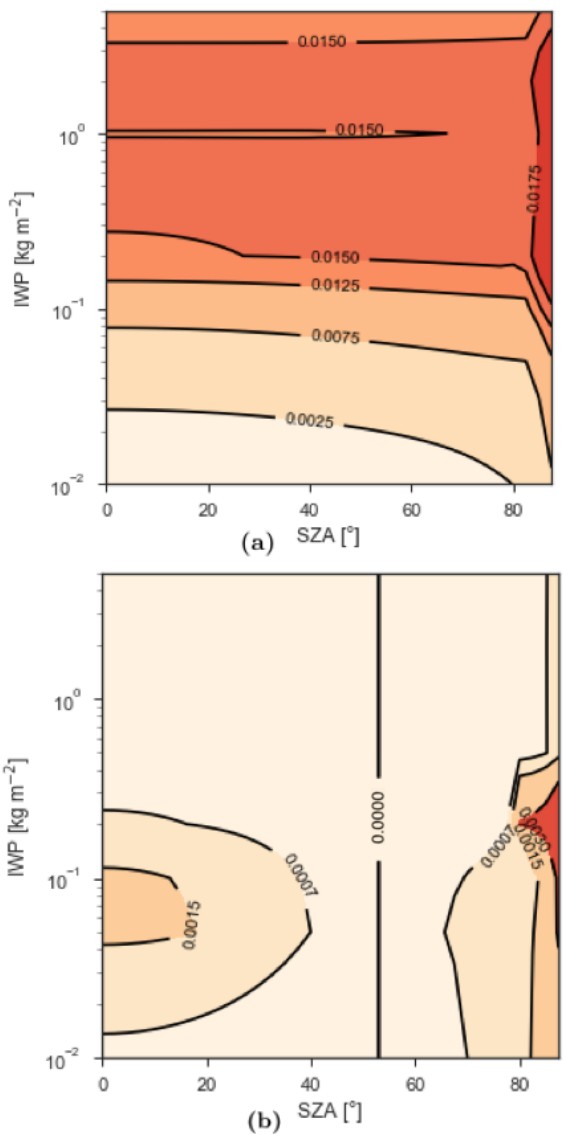

**Figure 6.** Weighted RMSE as a function of IWP and SZA, for LWP = 0.1 kg m$^{-2}$, (**a**) if the dependency of RWs to IWP and SZA is neglected, with RWs valid for IWP = 0 and LWP = 0.1 kg m$^{-2}$; (**b**) if the SZA dependency of RWs is neglected and taken at SZA = 53°.

Figure 7 summarizes the effect of LWP, IWP and SZA on the weighted RMSE of the derived narrowband albedos in more detail. In this figure, a LWP = 0.1 kg m$^{-2}$ and IWP = 0.1 and 1.0 kg m$^{-2}$ are used. The simplest and numerically fastest method is to convert the true effective spectral albedo into a broadband albedo using DISORT radiation, which is subsequently prescribed to each spectral band. This method therefore neglects sub-band variations. Figure 7 indicates that the broadband albedo method shows strong deviations, supporting Figure 3 that this method is not a viable option. Using RWs, but neglecting clouds, either with a direct SZA dependency, or with a fixed SZA, performs much better than the broadband albedo method, but still produces a considerable RMSE. If RWs depending on LWP and IWP are used, derived for a SZA of 53 degrees, the RMSE is negligible for all SZAs. For overcast conditions, RWs do, therefore, not need to depend on SZA. Finally, it is tested if a high-resolution dependency of the RW to e.g. IWP is required. For this aim, the RWs for only three IWP values are used, namely for IWP of 0.0, 0.5 and 2.0 kg m$^{-2}$, and RWs for intermediate IWPs are interpolated between these values. Figure 7 shows that in that case the RMSE increases, therefore, this low-resolution dependency is not an appropriate option. Similar results hold for limiting the LWP dependency to only a few values. In summary, RWs need to depend on LWP and IWP on sufficient cloud content resolution to capture the dependency of narrowband albedos to cloud cover. The SZA, on the other hand, can be safely neglected.

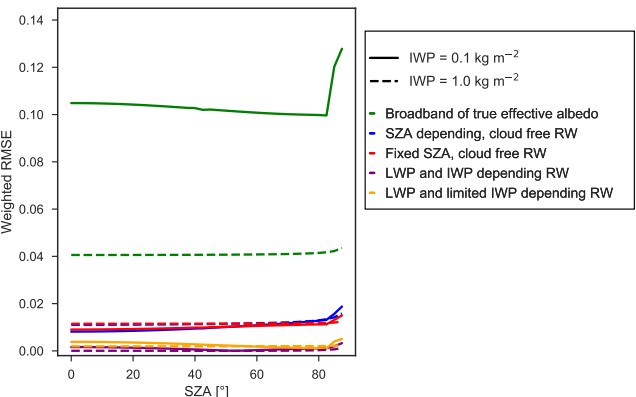

**Figure 7.** Weighted RMSE as a function of SZA with respect to RWs if LWP, IWP and SZA are all taken into account. The weighted RMSE is computed for different methods, keeping LWP = 0.1 kg m$^{-2}$ constant. The uninterrupted lines represent IWP = 0.1 kg m$^{-2}$ and the dashed lines IWP = 1.0 kg m$^{-2}$.

### 3.4 Impurities and snow profiles

Other factors controlling the narrowband albedo are the snow properties, of which impurities and snow grain size are the most important ones. Obviously, both are known to significantly alter the spectral albedo, as these variations are one of the primary reasons to start employing a narrowband albedo (Warren, 1982; Hoffer et al., 2006; Painter et al., 2009; Tedesco et al., 2016). However, an alteration of the spectral albedo does not necessarily coincide with a change of shape of the spectral albedo

profile, resulting in only minor RW changes. Consequently, the impact of snow properties on the RWs, which is expressed as the weighted albedo RMSE, is investigated.

Figure 8a shows schematically the RMSEs for clear-sky and overcast conditions explored in Sections 3.1 and 3.3, for different snow properties while using RWs depending on SZA for clear-sky diffuse, SZA and IWV for clear-sky direct, and LWP and IWP for overcast conditions, but defined for a fresh snowpack on top of melted snow of the previous year. For reference, the RMSEs that arise if a broadband albedo or invariant RWs were used, are also shown. In general, the weighted RMSE is low, i.e. < 0.01, for black carbon (BC) and HULIS. The albedo drops with impurity concentration, but the shape of the spectral albedo profile remains largely the same. Therefore, the computed RWs are similar to the RWs of the clean snowpack. This results in a low RMSE if the newly computed RWs are used to compute the narrowband albedos instead. Still, the RMSE increases with impurity concentration, while the spread decreases. The median RMSE is still low enough to be neglected safely, especially if it is compared with the errors that would arise if fixed RWs or a broadband albedo scheme are used. The effect of dust is not considered, because Mie scattering is not implemented in this version of TARTES. Finally, the impact of the snow grain radius and its vertical distribution is limited on the RWs that govern the narrowband albedo. In Figure 8a, the spread of RMSE of the considered snow profiles is high, indicating a large variability, but the median is low nonetheless. In general, a typical RACMO2 winter snow profile is more similar to the default snowpack than a typical RACMO2 summer snow profile that has experienced surface melt, resulting in larger RMSEs for summer conditions. Still, the typical error in the narrowband albedo for summer conditions is at most 0.01, and often lower and thus acceptable. The broadband albedo bias shows the same results (Figure 8b). A very low bias for both soot and HULIS is observed, while the bias for the summer snow profile is higher and positive, but still low (< 0.01) and acceptable. In conclusion, RWs depend mostly on radiation and not on snow profile states, resulting in a low RMSE and broadband albedo bias for variables impacting the snow profile. TARTES using RWs is therefore well capable to model the narrowband albedos for a wide range of snow profiles without including a dependency of RWs to the snow profile state.

## 3.5 Cloud properties

LWP and IWP are chosen to represent the effect of clouds on the RW. In addition, microphysical properties of clouds such as the cloud effective radius $r_e$ are known to impact the incoming radiation (Nielsen et al., 2014). We have chosen a realistic value of $r_e$, but in practice $r_e$ will vary for each instance. Although the potential effect of $r_e$ on the RWs is larger than BC and HULIS, it is still low (weighted RMSE < 0.01) for both clouds with small and large $r_e$, i.e. $r_{e,\text{ice}}$, $r_{e,\text{liquid}}$ = 15, 5 and 30, 30 μm respectively (Figure 8). These values for $r_e$ are on the lower and upper end of the probability range one could expect for the Arctic (King et al., 2004). Consequently, the typical weighted RMSE and bias is lower than indicated in Figure 8 and there is no need to make RWs dependent on $r_e$.

An alternative to the approach described in section 3.3 is the use of the cloud optical thickness $\tau$ instead of LWP and IWP to calculate RWs. This would be a valid approach if the spectral distribution is not altered considerably differently for ice clouds than for water clouds, as otherwise it would result in different RWs. Some differences between ice and liquid clouds are observed and are mostly caused by the various possible grain shapes and orientations of ice grains (King et al., 2004;

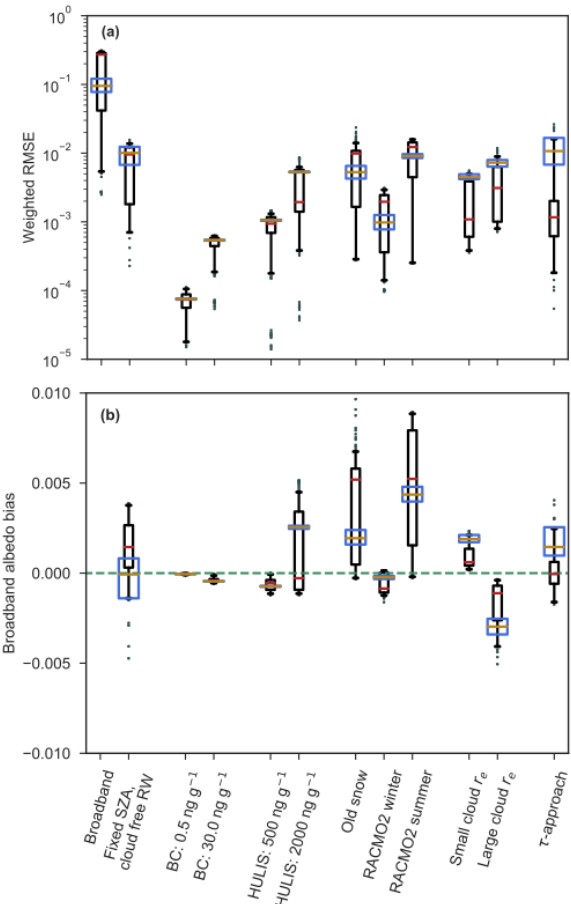

**Figure 8.** (**a**) Weighted RMSE of the broadband albedo method, fixed SZA, cloud free RW method, BC, HULIS and snow profiles. For each variable, the weighted RMSE is calculated as a function of SZA for clear-sky conditions for diffuse radiation, IWV and SZA for direct radiation, and as a function of SZA, LWP and IWP for overcast conditions. Together, they form an ensemble of atmospheric conditions, with the red line indicating the median, the box shows the 25th to 75th percentiles and the whiskers show the 5th to 95th percentiles. The dots show the outliers beyond the whiskers. The blue box shows the 25th to 75th percentiles for cloudy conditions if limited to LWP and IWP < 1.0 kg m$^{-2}$, with the dark orange line indicating the median. Low and high concentrations of BC and HULIS are considered. In addition, a hypothetical old snow profile is assessed, which consists of four layers with a density of 350, 400, 500 and 700 kg m$^{-3}$ and a SSA of 10, 5, 1 and 0.1 m$^2$ kg$^{-1}$. The weighed RMSE of a typical winter and summer snow profile are assessed as well. These profiles are extracted from RACMO2, and consist of many layers. The summer profile includes the impact of melt in the upper snow layers. The impact of cloud effective radius $r_e$ is evaluated for small and large values, i.e. $r_{e,\text{ice}}$, $r_{e,\text{liquid}}$ = 15, 5 and 30, 30 μm respectively. Finally, the $\tau$-approach is shown, as is described in section 3.5. Only cloudy conditions are considered for the cloud effective radius and $\tau$-approach. (**b**) The broadband albedo bias for the same variables. The bias of the broadband albedo method is 0 by construct and therefore left out.

Wyser and Yang, 1998). Still, a method using $\tau$ could be used if the uncertainty is small enough, but a choice regarding what type of clouds to compute $\tau$ for, i.e. ice clouds, liquid water clouds or a combination, and its cloud properties has to be made nevertheless and will inevitably lead to uncertainties. We tested this "$\tau$-approach", hence derived RWs as a function of $\tau$ for pure ice clouds, and linearly interpolated RWs for a given $\tau$ for liquid water clouds or a combination of liquid water and ice clouds. The approach performs reasonably well (Figure 8), but the spread is large. If the statistical analysis of the "$\tau$-approach"

is limited to common LWPs and IWPs in the Arctic ($< 1.0$ kg m$^{-2}$, see Figure 9), the RMSE is rather high (blue box and dark orange median in Figure 8), especially compared to the other parameters considered. Therefore, we decided not to use the cloud optical thickness as leading parameter to compute RWs.

## 4   Numerical implementation

In section 3, it has been shown that SZA for clear-sky diffuse, SZA and IWV for clear-sky direct, and LWP and IWP for

overcast conditions, are the most relevant factors controlling the RWs that are required to run TARTES in narrowband mode. Hence, three lookup tables were derived for each band, using the default aerosol loading and snow profile. The first lookup table is a two-dimensional table for overcast conditions, as the RWs depend on LWP and IWP. Similarly, the second lookup table depends on SZA and IWV for clear-sky direct. This lookup table varies between 0.5 and 40 kg m$^{-2}$ for IWV in ten steps, with a smaller interval between 0.5 and 10 kg m$^{-2}$. The third lookup table depends only on SZA for diffuse radiation for clear-

sky conditions. As an example, the lookup tables for cloudy conditions and clear-sky diffuse for band 6, which ranges between 778 and 1242 nm, are shown in Figure 9. The sub-band spectral shift due to clouds and its albedo effect (section 3.3) is clearly visible in a shortening of the RW for high LWP and IWP, leading to a higher narrowband albedo. Linear interpolation between RWs is used for conditions between the provided values of the lookup tables. For very thin clouds (LWP $< 0.05$ kg m$^{-2}$and IWP $< 0.01$ kg m$^{-2}$), a part of the irradiance is direct. This is modeled by linear interpolation, assuming that all irradiance is

clear-sky for LWP and IWP = 0.0, which includes a direct and diffuse part following the direct fraction illustrated in Figure 9, and completely total-sky for LWP = 0.05 kg m$^{-2}$ and IWP = 0.01 kg m$^{-2}$. The interpolated RWs are then handed over to TARTES to compute a narrowband albedo.

On some occasions, such as a thick cloud cover or a high SZA, DISORT computes no incoming solar radiation for some bands. As a result, a RW cannot be computed. These missing RWs were filled with the valid RWs of the most similar conditions.

The lookup table of Figure 9 contains high values of IWP and LWP to allow RACMO2 to be run for lower latitudes and to ensure that RWs are always calculated, even if RACMO2 would produce unusually thick clouds. The albedo for direct radiation for clear-sky conditions is computed in a similar way, but the lookup table depends on SZA and IWV instead of LWP and IWP.

Finally, TARTES evaluates the albedo for direct and diffuse irradiance, while RRTM$_{sw}$ in RACMO2 is run for clear-sky and total-sky conditions. Therefore, the lookup tables for clear-sky conditions is extended with the direct to diffuse irradiance ratio

as function of the SZA, as derived using DISORT. Hence, two narrowband albedos have to be computed for all 12 bands for (near) cloud-free atmospheric conditions, one for direct and one for diffuse irradiance. One call to TARTES for each band is sufficient if the cloud fraction is 1 and either the LWP or IWP is larger than 0.05 or 0.01 kg m$^{-2}$, respectively. For fractional

cloud cover or if LWP and IWP are lower than 0.05 and 0.01 kg m$^{-2}$, respectively, TARTES is called upon for both clear-sky

and total-sky conditions, which results in three calls to TARTES. If LWP and IWP are 0.0 kg m$^{-2}$, clear-sky conditions equal

total-sky conditions, resulting in two calls to TARTES.

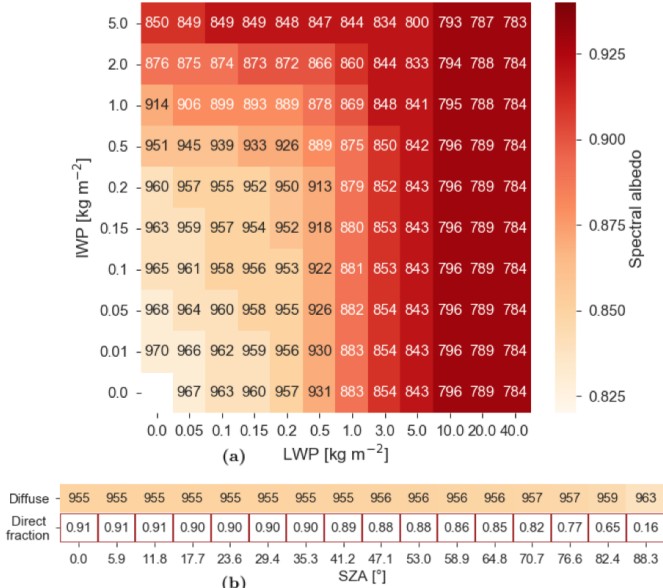

**Figure 9.** Final lookup table with RWs for band 6. Colours express the corresponding albedo for the default snowpack. (**a**) The two-dimensional table for overcast conditions, depending on IWP and LWP, and (**b**) clear-sky conditions table for diffuse radiation and the fraction of incoming radiation that is direct as a function of SZA. The white square at LWP, IWP = 0.0 kg m$^{-2}$ of the overcast table in (**a**) indicates the conditions when the clear-sky table of (**b**) is required. The lookup table for direct radiation is not shown, but is similar to (**a**) with SZA and IWV as dimensions.

## 5   Offline comparison with broadband albedo models

In order to assess the difference between TARTES, run in narrowband mode, with more traditional approaches, we compare

here the albedos derived with TARTES using SNOWBAL with the parameterizations of Gardner and Sharp (2010), GS, which

is also the default scheme in RACMO2, and Kuipers Munneke et al. (2011), PKM. All results presented here use the atmo-

spheric conditions, narrowband downwelling shortwave fluxes and snow profiles computed by RACMO2. Figure 10 shows

examples for winter (**a-e**) and summer (**f-j**), respectively. The winter snow albedo (**a**) is generally high due to slower snow

metamorphism and limited melt. During summer (**f**), melt events and subsequent refreezing alters the internal structure of the

snowpack, resulting in snow compaction and decreasing SSA. Furthermore, glacial ice with low albedo is exposed in the abla-

tion zone along the western margin. The domain averaged albedo of TARTES is lower than RACMO2, both in winter (**b**) and in

summer conditions (**g**). A significant part of the differences between TARTES and RACMO2 are due to the fact that RACMO2

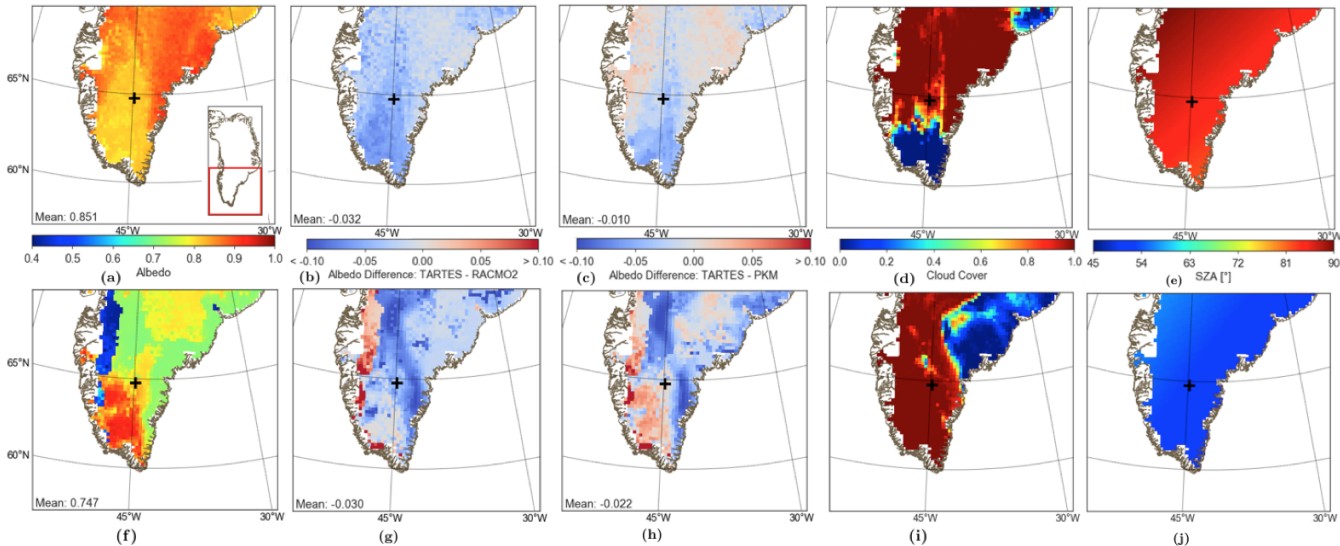

**Figure 10.** (**a, f**) Broadband albedo computed by TARTES using RACMO2 output for South Greenland during 15 February 2007, 12:00 UTC (upper row, 9:00 local time) and 15 July 2007, 12:00 UTC (lower row, 9:00 local time), (**b, g**) albedo difference between TARTES - RACMO2, (**c, h**) albedo difference between TARTES - PKM, (**d, i**) cloud cover, (**e, j**) and SZA. For (**a - c**) and (**f - g**), the domain averaged mean is indicated in the lower left corner. A plus symbol indicates the location of a time series analysis.

only considers the top two layers of the snowpack, regardless of the thickness. The differences between TARTES and PKM are generally less (**c, h**), leading us to conclude that the missing impact of deeper layers in RACMO2 leads to overestimated albedos. Still, large differences remain between TARTES and PKM. For fully overcast conditions (**d, i**), the differences are primarily due to the spectral variable radiation penetration into the snowpack. In winter, when the snow grain size gradually increases in the snowpack, TARTES estimates generally higher albedos than PKM for large regions of the ice sheet. For the specific summer situation of (**f - j**) that includes previously melted snow covered by freshly fallen snow, TARTES estimates lower albedos than PKM.

In the winter example, when the SZA is large (**e**) compared to the summer example (**j**), TARTES calculates a substantially lower albedo than RACMO2 and PKM for the southern tip of Greenland, which is cloud-free at that specific moment (**d**). This difference can be explained by the fact that two compensating effects occur for large SZA during clear-sky conditions. Firstly, a large SZA results in photons being more likely to scatter out of the snow, resulting in a reduced penetration and an increased albedo. Moreover, photons travel a longer path through the upper fresh snow layers before reaching older layers with a reduced SSA, which also increases albedo. Secondly, Rayleigh scattering in the atmosphere causes a spectral shift towards larger wavelengths, for which the albedo of snow is relatively low (Warren and Wiscombe, 1980; Ackermann et al., 2006; Warren et al., 2006; Gardner and Sharp, 2010). This spectral shift is not or not sufficiently included in PKM and RACMO2 (Figure 11). For high SZA, DISORT models a clear spectral shift towards longer wavelengths, limiting the increase of the broadband albedo. If this effect is left out (black dashed line) the broadband albedo is much higher. Hence, the difference

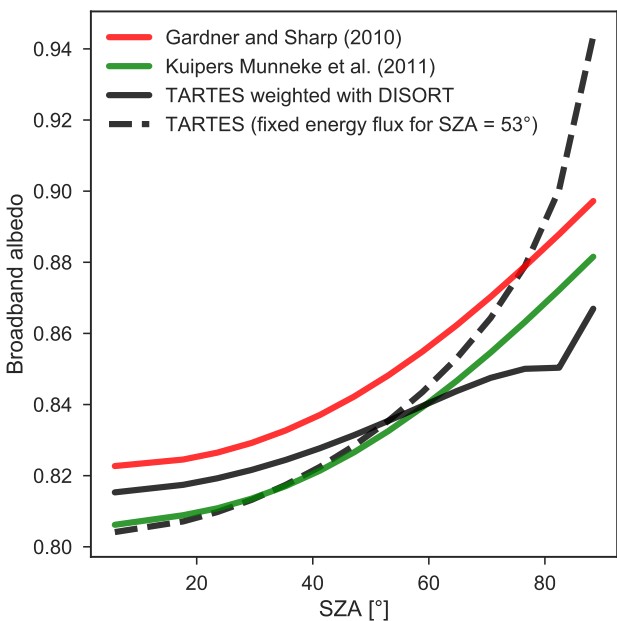

**Figure 11.** Example broadband albedo as a function of wavelength for the parameterization of Gardner and Sharp (2010), Kuipers Munneke et al. (2011) and TARTES. The spectral albedo of TARTES is weighted with energy fluxes derived with DISORT (in black, solid line) or with the energy fluxes valid for a SZA of 53° (black, dashed line) to compute a broadband albedo. The surface pressure in libRadtran is set to 900 hPa.

between the black solid and dashed line indicates this albedo decrease is not induced by the RW-approach, but by general red-shift in the incoming radiation. A similar effect does not occur for overcast conditions. Clouds preferentially absorb light
in the IR part of the spectrum, nullifying the spectral shift induced by atmospheric Rayleigh scattering.

    Figure 12 shows a time series for 2007 at 15:00 UTC (12:00 local time) at the location indicated in Figure 10. One data point per day is shown to remove the daily cycle in insolation and albedo, which would clutter the graphs. As for most of the ice sheet, clouds frequently cover the sky fully, but these clouds are often thin. Furthermore, this site experienced in this year three rather short melt periods. In line with Figures 10 and 11, TARTES and PKM systematically estimate lower albedos than
RACMO2 (Figures 12b - d). During the winter months and with overcast conditions, the albedo difference between TARTES and PKM is limited, indicating the importance of the deeper snow layers on the snow albedo. However, for clear-sky conditions with a high SZA, TARTES systematically calculates lower albedos than RACMO2 and PKM (Figure 12b - f), which coincides with a spectral shift towards IR radiation (Figure 12a). Hence, the spectral shift during clear-sky conditions and high SZA is an important factor for the snow albedo. Large albedo differences are also modeled in the period after the second melt event.
This second melt event leads to a strong metamorphism of snow and the removal of the high SSA top snow layers (Figure 12h). As a result, the albedo drops to approximately 0.75. After this melt event, a new fresh snow layer is formed in early July.

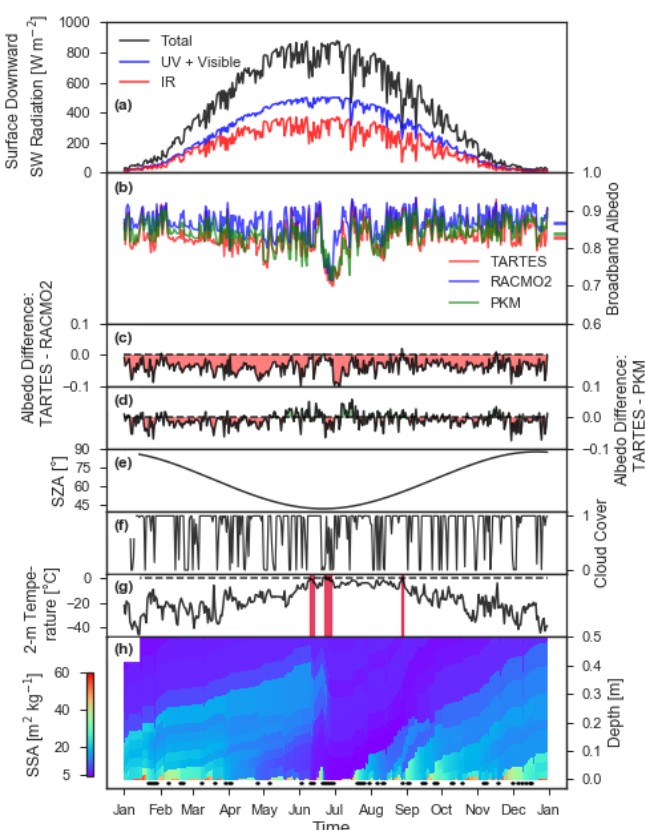

**Figure 12.** Time series for 2007, 15:00 UTC (12:00 local time) at the location indicated by the plus sign in Figure 10. (**a**) The surface downward shortwave radiation, with the total flux split into an IR and UV plus visible radiation part; (**b**) broadband albedo computed by TARTES weighted with DISORT, RACMO2 and PKM, the mean albedo is indicated by the horizontal bars; (**c**) the albedo difference TARTES - RACMO2 and (**d**) TARTES - PKM, with red indicating a negative and green a positive difference; (**e**) solar zenith angle; (**f**) cloud cover; (**g**) 2-m temperature, with red bars indicating melt days; (**h**) specific surface area of the snow layers as a function of depth, with the horizontal black bars at the bottom indicating moments without a top fresh snow layer.

This fresh snow layer is initially very thin, less than a millimeter, but nonetheless strongly raises the albedo of RACMO2. This limited effect of a very thin snow layer is better represented by TARTES and PKM, where the albedo only gradually recovers from the second melt event. Still, this recovery is modeled differently by PKM and TARTES; TARTES has a faster recovery than PKM. The wavelength-dependent effect of radiation penetration is thus important for a very inhomogeneous snowpack. As is expected by Kuipers Munneke et al. (2011), this effect diminishes when the new fresh snow layer becomes thicker and the underlying old snow profile becomes less important. Note that the differences between TARTES and PKM are typically larger than the typical errors, i.e. < 0.01, induced by using narrowband irradiance and RWs compared to the full spectral model (Figure 8b).

# 6 Discussion

We have presented the SNOWBAL module to couple a narrowband regional climate shortwave radiation model with a spectral albedo model by using representative wavelengths. After calculating a RW, which depends on the SZA for clear-sky conditions and on LWP and IWP for overcast conditions, a narrowband albedo can be computed for that RW. We have shown that the albedo errors that arise due to the use of this method are relatively small ($< 0.01$) The uncertainties induced by other modeled physical properties that influence the albedo, like SSA and impurity content, are often larger.

Impurities are shown to have a limited effect on the RW and have been neglected in the time series analysis. Still, impurities will impact the albedo. However, RACMO2 lacks a sophisticated impurity scheme and impurities are parameterized differently compared to RACMO2 and PKM, thus making it difficult to assess the effect of impurities on the albedo using TARTES. In future work, vertical profiles of impurities will be included and assessed in more detail.

Small errors may also arise as the effect of snow grain shape, which is implemented in TARTES, is not taken into account as RACMO2 does not model the grain shape evolution. In TARTES, the effect of grain shape on the radiation penetration is parameterized using the absorption enhancement parameter $B$, which signifies the enhanced absorption due to the change of the photon path within a grain by internal reflections, and a geometrical optics asymmetry factor $g^G$. Libois et al. (2014) assess that $B$ is best set to represent hexagonal plates, which then yields $B = 1.6$, but that the ratio $B/(1 - g^G)$ should equal that of spheres, which yields $g^G = 0.72$. Libois et al. (2014) also demonstrate that uncertainties in $B$ and $g^G$ are irrelevant compared to uncertainties in SSA. Therefore, it is not deemed necessary to include a sophisticated grain shape parameterization in RACMO2 nor to determine the effect of grain shape on RWs. Other properties can possibly affect the spectral albedo of snow, e.g. cloud top height, but their effect is deemed negligible compared to the other known uncertainties.

Using the cloud optical thickness instead of LWP and IWP to calculate RWs does not have the desired result, as a distinction between ice and liquid water clouds still have to be made. Moreover, the quality of the liquid cloud optical thickness parameterization by Slingo (1989) in the IFS part of the ECMWF model version used in RACMO2 is limited and outdated (Hogan and Bozzo, 2018; Nielsen et al., 2014), and is updated in later iterations of the ECMWF model, but not available yet for RACMO2. In addition, the ice cloud parameterization by Fu (1996), which is used in RACMO2, is not so reliable for thicker clouds above surfaces with a high albedo (Nielsen et al., 2014). Therefore, the use of cloud optical thickness in RACMO2 to determine RWs would result in an additional uncertainty on top of the described uncertainties in section 3.5. Hence, the option to use cloud optical thickness instead of LWP and IWP has been dismissed.

The comparison of TARTES with RACMO2 and PKM has been limited to one year for South Greenland, as this comparison aims to be a proof of concept. Indeed, many differences between the albedo parameterizations of Gardner and Sharp (2010) and Kuipers Munneke et al. (2011) have been observed. In a subsequent paper, TARTES implemented in RACMO2 will be thoroughly evaluated using in-situ measurements and remote sensing observations and its performance will be compared with other albedo schemes.

## 7 Conclusions

Energy fluxes in climate models are often provided in spectral bands. This study presents the simple, yet effective SNOWBAL module to couple a spectral snow albedo model with narrowband atmospheric radiation schemes. For this coupling, a representative wavelength is computed for each spectral band to incorporate sub-band irradiance and albedo variations. Ideally, the RW includes the effect of many physical processes, but numerical constraints limit this to the most important variables. For clear-sky conditions, the SZA for both direct and diffuse irradiance, IWV for direct irradiance, and the ratio between direct and diffuse are the primary physical properties to consider for the computation of RWs. For overcast conditions, considering the LWP and IWP is sufficient to capture most sub-band variations associated with cloud cover. Hence, RWs can be parameterized with three lookup tables for each spectral band, the first for overcast conditions, providing RWs as a function of LWP and IWP, the second for clear-sky direct as a function of SZA and IWV, the third for clear-sky diffuse, providing RWs as function of SZA. In addition, the fraction of direct radiation with respect to the total irradiance has to be used.

We apply the new method to couple the spectral radiative transfer in snow model TARTES with the atmospheric radiation scheme of the ECMWF model in RACMO2.3p2, and compare the narrowband albedo with broadband albedo parameterizations of Gardner and Sharp (2010), which is embedded in RACMO2, and the multi-layered broadband albedo scheme of Kuipers Munneke et al. (2011), which essentially uses GS for multiple layers. A model intercomparison for South Greenland in 2007 shows that the domain averaged broadband albedo computed by TARTES is lower than RACMO2. A large part of this discrepancy is because RACMO2 only considers the top two layers of the snowpack, regardless of thickness. The differences between TARTES and PKM are smaller, illustrating the relevance of radiation penetration. For clear-sky conditions during winter, i.e. large SZA, we show that the spectral shift towards larger wavelengths has a substantial impact on the albedo, resulting in an albedo decrease. This effect is not taken into account by either RACMO2 and PKM, resulting in an albedo overestimation of these parameterizations with respect to TARTES. After a melt event, fresh snow layers gradually reform. As a consequence, the albedo of RACMO2 quickly rises to high values, because it only considers the top two layers. On the other hand, TARTES and PKM recover more gradually. Still, TARTES and PKM recover at a different rate, which reveals the relevance of the wavelength-dependent effect of radiation penetration for an inhomogeneous snowpack.

To conclude, a coupled spectral albedo model potentially leads to improved albedo estimates. This study discusses the implementation strategy and provides a proof of concept. In forthcoming publications, the performance gain introduced by using TARTES in RACMO2 will be evaluated using in-situ measurements and remote sensing observations.

*Code and data availability.* The SNOWBAL module v1.2 written in Python, alongside a code to produce a time series of RACMO2 using SNOWBAL, are supplemented to this paper. TARTES is freely avaiabable on the website: http://pp.ige-grenoble.fr/pageperso/picardgh/tartes/. The libRadtran software package including DISORT is also freely available: http://www.libradtran.org/doku.php. RACMO2.3p2 model data used in this paper can be found here: https://doi.org/10.5281/zenodo.1468647.

*Author contributions.* CTVD, WJB and MB started this project, decided which spectral snow albedo model to use and interpreted the results. CTVD performed the model simulations, creation of SNOWBAL, comparisons and led the writing of the manuscript. QL and GP provided information and feedback regarding TARTES. All authors contributed to discussions on the manuscript.

*Competing interests.* The authors declare that they have no conflict of interest.

*Acknowledgements.* We acknowledge financial support from the Netherlands Organization for Scientific Research (NWO) and the ECMWF for computational time on their supercomputers.

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
