# Peer review of "A module to convert spectral to narrowband snow albedo for use in climate models: SNOWBAL v1.2"

_Geoscientific Model Development, 2018_

## Referee Comment (RC1) · Anonymous Referee #1 · 3 Dec 2018

This paper presents a novel method for coupling the snow spectral albedo model TARTES to the regional climate model RACMO using a representative wavelength approach under both direct and diffuse illumination conditions, with the goal of improving the albedo scheme and hence energy budget simulation of the Greenland Ice Sheet. The method revolves around the generation of lookup tables which define the representative wavelength of each RACMO narrow-band as a function of illumination conditions. This makes it possible to run TARTES within RACMO, with two principal benefits: (1) significant improvements in the computation of the snow albedo within RACMO compared to the existing broadband approach, and (2) computational efficiency maintained by running TARTES only once for each RACMO narrow-band - using the representative

wavelength - as opposed to for the entire shortwave part of the spectrum. The study shows that very large differences can be found between the broadband and spectral approaches and hence concludes that treating snow albedo spectrally is critical to a good energy budget simulation.

I consider this study to be timely and important for two reasons. First, it clearly illustrates that existing broadband albedo calculation approaches introduce significant uncertainties in energy budget calculations because they are unable to explicitly account for spectrally dependent processes including clouds and snow metamorphism. Second, given the recent literature which seeks to understand the impact of cloud cover upon GrIS runoff during the coming century (e.g. van Tricht et al., 2016, Hofer et al., 2017), it is timely to consider whether the models used to investigate cloud cover are capable of adequately resolving their impact upon albedo and hence runoff. To my understanding, the methods chosen are appropriate to answer these kinds of questions. The study combines existing mature models together through its elegant computation of lookup tables for representative wavelengths.

The model code and a user manual are provided. I have not tried to run the code as there are a number of dependencies that I would need to install first. The code is reasonably well commented and clear and I would not anticipate experiencing problems with its use as a third-party user.

I consider the manuscript to be in excellent shape. The figures and tables are all of good quality. I have only a few, relatively minor comments:

* A more appropriate reference than Dumont et al.(2014) for P1, L21-23 should be sought. The Dumont paper was really about the potential impact of impurities on snow albedo and hence melt, and subsequent research suggests that its conclusions were not correct (Polashenski et al., 2015). * The study mentions that a spectral approach is important for examining the impact of impurities upon snow albedo in the abstract, but later on (page 14, L13-17), states that the effect of dust is not considered as TARTES

uses the delta-Eddington approach (P5, L7) as opposed to Mie scattering. This seems a fundamental issue and so I wonder if the study would benefit from an additional paragraph, perhaps in the introduction, which outlines the different optical approaches and what they permit in terms of albedo modelling. Reference to Cook et al. (2017) may be useful here. * The text can be tightened in places - many sentences are very wordy and could be simplified without loss of meaning.

References not in the text:

van Tricht et al. (2016), Nat. Commun., 7, 10266. Hofer et al. (2017), Sci. Adv., 3, e1700584. Polashenksi et al. (2015), Geophys. Res. Lett., 10.1002/2016GL065912. Cook et al. (2017), The Cryosphere, 11, 2611-2632.
* * *

---

## Referee Comment (RC2) · Kokhanovsky (Referee) · 18 Dec 2018

The paper is aimed at the description of the module to convert spectral to narrowband snow albedo for use in climate models.

I have the following comments:

p.3 TARTES is based on a simple approximation. Therefore, it should not be a problem to run it for multiple wavelengths. Please, give the estimation of time needed to produce the spectral albedo as shown in Fig.1. Please, explain in the paper why you need to make the calculations of snow albedo at hundreds to thousands wavelength using

TARTES. I guess, the spectral snow albedo as shown in Fig.1 can be calculated on a fixed spectral grid (say, 30-50 wavelengths) and supplemented with a simple interpolation routine to derive it on yet another grid needed for the integration with the solar irradiance as shown in Fig.1. Could you show errors of this simple approach suggested by me in the paper.

p.5 Please, change: 'geometric asymmetry parameter' to 'geometrical optics asymmetry parameter' (see also p.21). To be more clear, please, acknowledge in the paper that the total asymmetry parameter $g=(1+g\_G )/2$ for nonabsorbing particles. Please, explain in the paper why $B/(1-g\_G )$ must be equal to the corresponding value for spheres.

p.19, Fig.11, TARTES albedo drops at high SZA. Please, explain the reason for this. I guess, this is not a correct behavior. Are you aware of experimental results which confirm such a drop in albedo? I would suggest to make a plot of BBA as function of cos(SZA).

p.21, Please, change 'assymetry' to 'asymmetry'

––––––––––––––––––––––

---

## Referee Comment (RC3) · Anonymous Referee #3 · 27 Dec 2018

This is a very ambitious paper that addresses a very important issue in atmospheric modelling of properly coupling atmospheric radiative transfer computation to the radiative transfer computations of a snowpack. A main issue at present is that these two forms of computations have been developed separately, and the different spectral bands are used in each type of model. The authors aim to combine the RRTM_SW atmospheric radiative transfer model, with 14 shortwave (SW) spectral bands, and the TARTES snowpack radiative transfer model, that includes more than 100 spectral bands. The method is to find representative wavelengths (RWs) from TARTES for each of the 14 spectral bands in RRTM_SW. Thereby, the use of TARTES can be made more efficient. The RWs are chosen based on multi-spectral DISORT/libRadtran

simulations combined with TARTES simulations.

The paper is generally well-written and -structured.

Major issues:

The authors claim to have developed their scheme, so that the relative RMSE of this for each spectral band is less than 0.01 (or 1%) relative to the "fully spectral DISORT and TARTES calculations." This can easily be read as if the general error is less than 1%, however, that depends on the accuracy of the performed DISORT/TARTES calculations! Here the DISORT computations are run with the libRadtran script library, which is easy to use but which easily can be used incorrectly. Specifically, I have the following issues:

1a) DISORT is here run with 6 streams. Have you tested if that is enough to get the desired accuracy? If not, you should do so! Quote from Stamnes et al. (2000): "For strongly forward-peaked phase functions it is difficult to get accurate intensities with fewer than 16 streams, and even with 16 streams accuracy can be poor at some angles. Thus, careful users have been forced to use 32 or even 64 streams to be sure of getting 1% accuracy" Stamnes et al. here speak of intensities and not fluxes for which less streams are required, however, I expect that 6 streams are too little also to obtain very accurate fluxes. Tests should be done particularly for high solar zenith angles (SZAs), as these often occur in Greenland.

1b) In the DISORT simulations the surface broadband albedo is set to 0.5. In the supplementary scripts, it can be seen that this is done regardless of wavelength with the "albedo" input option in libRadtran. Here, the "albedo_file" input option should have been used, in which spectral albedos can be specified. In order to make the atmospheric and snowpack radiative transfer computations consistent, the TARTES spectral albedos should be used in this albedo file. As shown by for instance Nielsen et al. (GMD, 2014), the downward fluxes at the surface are not independent of the albedo. Given the complex variations of spectral irradiances and albedos shown in Fig. 1, it

seems important to run DISORT with the TARTES albedos. Fig. 1 is a very illustrative figure by the way! An even better representation of the surface reflectance could be obtained by running coupled DISORT simulations for both the snowpack and the atmosphere. This can be done by adding the spectral inherent optical properties of the layers of the snowpack as the lowest model layers in DISORT. In this way the full BRDF of the snow surface will be properly represented and coupled with the atmospheric simulations, which cannot be done with the two-stream TARTES simulations.

1c) The "subartic winter" atmospheric profile is used. The reference describing the details of this is missing and should be added. I assume that this is one of the AFGL standard atmospheres of Anderson et al. (1986). Additionally, the "rural aerosols" of Shettle (1990) are used. How representative are these profiles for Greenland? Could typical atmospheric profile data from the CAMS reanalysis be used in stead? The clear sky spectrum can change quite a lot depending on the gasses and aerosols assumed to be present. You should mention this uncertainty in the method chosen.

1d) When inputting liquid and ice clouds to DISORT you assume these to have effective/equivalent radii of 10 $\mu$m and 20 $\mu$m, respectively. Here the former number is reasonable, but 20 $\mu$m is a very low number for typical ice clouds, where I would suggest using 50 $\mu$m in stead. Also, you make these look-up table values a function of the cloud optical thickness rather than the cloud liquid water path (LWP) and ice water path (IWP). Since the cloud optical thickness is proportional to LWP+IWP and approximately inversely proportional to the effective/equivalent radii, similar relative changes in these cloud properties cause similar changes to the cloud optical thickness.

1e) In the DISORT experiments a range of cloud ice water path of up to 5 kg/m$^2$ is simulated. This is at least 10x more than a realistic maximum value for clouds over Greenland. Cloud liquid water paths of up 40 kg/m$^2$ are also simulated. This is also an order of magnitude higher that cloud water paths that can occur even in the tropics. I suggest that the simulations are done for more realistic ranges. Also, the results shown in Fig. 7 are run for a LWP of 0.5 kg/m$^2$. Is that a typical LWP for clouds over

Greenland? Please update your experiments to more typical values.

1f) In the supplementary scripts it can be seen that the rte_solver disort is used in the libRadtran/DISORT experiments. Here, the rte_solver sdisort (Dahlback & Stamnes 1991) should be used in stead. The regular disort/disort2 solver is designed for a plane-parallel geometry, where the atmosphere curves. sdisort is a pseudo-spherical disort solver, which accounts for the atmospheric curvature. In particular for high SZAs using disort will cause errors. This is also likely to explain the discrepancies seen for high SZAs in Fig. 11.

-o-

2) Page 21, lines 14-15: "However, this version of the IFS code embedded in RACMO2 does not explicitly model any cloud content properties nor includes a parameterization of the optical depth." That is incorrect! Since you have not included the RACMO2 source code in your supplementary material, I cannot tell what "the IFS code embedded in RACMO2" entails, but I am very familiar with the IFS radiation scheme and SRTM as used in cy33r1. In this the cloud optical thickness is parameterized. In fact it is computed for each spectral band in each 3D model grid box.

3) Page 9, line 10: "The broadband albedo, which is for direct radiation close to 0.78 for most SZAs..." The broadband albedo of snow can be quite different from 0.78 depending on atmospheric and snow conditions. Please correct this line to reflect this!

-o-

Minor comments:

- Abstract, line 5: "... Integrated Forecast System atmospheric..." –> "... Integrated Forecast System (IFS) atmospheric..." Also, add the version number 33r1!

- Abstract, line 10: "... 14 spectral bands of the ECMWF shortwave..." –> "... 14 spectral bands of the IFS shortwave..."

- Page 1, lines 22-23: "... the melt-albedo feedback, e.g. Dumont et al. (2014)." –> "... the melt-albedo feedback (e.g. Dumont et al., 2014)."

- Page 2, line 4: "... solar angle" –> "... solar zenith angle"

- Page 2: You need to add spectral band definitions of what you mean, when you refer to "near-UV" and "near-IR".

- Page 2, lines 13-14: "... the ratio of upwards to downwards shortwave radiative flux integrated over the solar spectrum." –> "... the ratio of upwards to downwards shortwave radiative flux on a horizontal surface integrated over the solar spectrum." Here you should also add explanations of the direct ("black sky") albedo, which varies as a function of the SZA, and the diffuse ("white sky") albedo. Both of these are used as input variables to SRTM in the IFS radiation scheme.

- Page 2, lines 19-20: "For example, a buried dark impurity layer will only significantly affect near-UV albedo." This I disagree with. Water has minimal absorption at the UV/violet spectral boundary, and the absorption is also very low in the UV, blue and green parts of the spectrum. Thus, these parts are also significantly affected by underlying impurities.

- Page 2, line 21: "... the thermal regime" –> "... the snow heating rates"

- Page 2, line 30: The RRTM_SW (also known as SRTM in the IFS code) references are placed after "RACMO2" in this line. They should be moved back to where RRTM_SW is referred to!

- Page 4, lines 1-2: "RRTM_SW computes flux profiles for clear-sky and total-sky conditions on hourly intervals." –> "RRTM_SW computes instantaneous flux profiles for clear-sky and total-sky conditions."

- Page 4, lines 14-15: "... specific surface area (SSA)..." This is the same acronym that is used for single-scattering albedo, an essential radiative transfer variable, which can be confusing. You should consider using something else.

- Page 5: lines 31-32: "The effective droplet radius for ice and water clouds ..., which is a realistic radius for clouds" –> "The effective droplet radius for ice and water clouds ..., which are realistic radii for clouds"

- Page 10, lines 20-21: See my comment 1b above.

- Figure 12: This is a very nice figure, however, it is very difficult to distinguish cloud covers of 0 and 1. Please expand this part of the figure, so that these data are not hidden by the graph axes!
* * *

---

## Author Comment (AC1) · 6 Feb 2019

Referee comment response on the manuscript: A module to convert spectral to narrowband snow albedo for use in climate models: SNOWBAL v1.0 by C.T. van Dalum et al.

We would thank the reviewers for their constructive comments that have improved the accuracy of the calculations and the clarity of the paper. For an easier version to read of the referee comment response, see the supplemented pdf file, where the following colors are used: in black the comment, in orange the response, in blue the changes.

[Figure]

Review #1

Comment 1: A more appropriate reference than Dumont et al.(2014) for P1, L21-23 should be sought. The Dumont paper was really about the potential impact of impurities on snow albedo and hence melt, and subsequent research suggests that its conclusions were not correct (Polashenski et al., 2015).

You are right and we have changed the reference to Van As et al, 2013.

Comment 2: The study mentions that a spectral approach is important for examining the impact of impurities upon snow albedo in the abstract, but later on (page 14, L13-17), states that the effect of dust is not considered as TARTES uses the delta-Eddington approach (P5, L7) as opposed to Mie scattering. This seems a fundamental issue and so I wonder if the study would benefit from an additional paragraph, perhaps in the introduction, which outlines the different optical approaches and what they permit in terms of albedo modelling. Reference to Cook et al. (2017) may be useful here.

We have added extra explanation in the introduction to include what you have mentioned. Page 2: Impurities mostly affect the reflectivity for near-UV and visible light, while snow metamorphism mostly affect the reflectivity for near- IR light (Tedesco et al., 2016). The grain radius of impurities determines the scattering regime. The typical grain radius of soot and humic-like substances (HULIS) are small compared to short-wave wavelengths, while the typical grain radius of dust is not small. Consequently, an albedo model has to be compatible for Rayleigh scattering to incorporate soot and HULIS, and for Mie theory for dust and biological material (Tegen and Lacis, 1996; Cook et al., 2017). Page 15: The effect of dust is not considered, because Mie scattering is not implemented in this version of TARTES  

Review #2

Comment 1: TARTES is based on a simple approximation. Therefore, it should not be a problem to run it for multiple wavelengths. Please, give the estimation of time needed

to produce the spectral albedo as shown in Fig.1. Please, explain in the paper why you need to make the calculations of snow albedo at hundreds to thousands wavelength using TARTES. I guess, the spectral snow albedo as shown in Fig.1 can be calculated on a fixed spectral grid (say, 30-50 wavelengths) and supplemented with a simple interpolation routine to derive it on yet another grid needed for the integration with the solar irradiance as shown in Fig.1. Could you show errors of this simple approach suggested by me in the paper.

It would likely be possible to reconstruct the spectral albedo of a snowpack for a certain SZA with a limited number of wavelengths, as suggested. Within the SNOWBAL framework, calculation time is of lesser importance. Therefore, we did not see the necessity to optimize the spectral albedo calculations. The time it takes for TARTES to compute an albedo increases quite significantly with the number of wavelengths taken. For example, for a 50 layer snowpack, the Python version of TARTES only takes 0.007 seconds to compute a spectral albedo for 12 wavelengths, while it takes about 0.268 seconds for 1000 wavelengths, which is about 38 times longer.

The reason not to adapt this approach for modeling albedo in an RCM is that an albedo curve can only be appropriately used if the spectral distribution of incoming irradiance is available with sufficient spectral detail too. And that is not the case as RACMO provides the irradiance in 14 rather wide bands. Hence, even with an efficiently derived fully spectral snow albedo, it would not be possible to estimate the snow albedo accurately within RACMO as sub-band energy fluxes are essential but unavailable. Therefore, we discarded this idea in an early phase of the project. Page 6: this would lead to a significant numerical burden, although it is likely possible to parameterize this spectral curve using in the order of thirty well-chosen spectral albedos. Page 6: However, RACMO2 does not compute sub-band energy fluxes. Hence, even with an efficiently derived fully spectral snow albedo or with smaller spectral bands, it would not be possible to estimate the snow albedo accurately within RACMO2 as sub-band energy fluxes are essential, but unavailable. Therefore, we discard this approach.

Comment 2: Please, change: 'geometric asymmetry parameter' to 'geometrical optics asymmetry parameter' (see also p.21). To be more clear, please, acknowledge in the paper that the total asymmetry parameter g=(1+g_G)/2 for nonabsorbing particles. Please, explain in the paper why B/(1-g_G) must be equal to the corresponding value for spheres.

In the ART formalism, snow optical properties depend on snow SSA, g, B and snow density (all single scattering properties can be derived from these properties). Practically, people have been using spheres to represent snow. The parameters B and g can be computed for spheres. This approach proved quite successful for albedo computations (which is why it is widespread), but much less for transmittance or penetration depth simulations. Albedo depends on B/(1-gG) while penetration depth depends on B*(1-gG). Libois et al., (2014) developed a method to determine B, and demonstrated at the same time that g cannot be determined based on optical measurements - it is coupled to SSA. This means g must be assumed somehow. The relative success of spheres means that any shape such that B/(1-gG) equals that of spheres should be quite efficient for albedo simulations. Hence the best estimate of g would be such that B/(1-gG) equals the value for spheres. Page 5: . . .and grain shape is determined by a geometrical optics asymmetry parameter gG, with the total asymmetry parameter g = 1/2 (1+gG) for non-absorbing particles, and an absorption enhancement parameter B. Page 5: The parameters B and g can be computed for spheres and prove to be quite successful for albedo calculations (Gallet et al., 2009, Grenfell and Warren, 1999), but much less for transmittance or penetration depth simulations. Libois et al., (2014) demonstrate that g cannot be determined based on optical measurements, because it is coupled to SSA, and must be assumed somehow. The relative success of spheres for albedo calculations, which depends on B/(1-gG), means that any shape such that B/(1-gG) equals that of spheres should be quite efficient for albedo simulations. Hence, the best estimate of g would be such that B/(1-gG) equals the value for spheres.

Comment 3: Fig.11, TARTES albedo drops at high SZA. Please, explain the reason for

this. I guess, this is not a correct behavior. Are you aware of experimental results which confirm such a drop in albedo? I would suggest to make a plot of BBA as function of cos(SZA).

To clarify what we meant, we have added an extra line in Figure 11 that shows the albedo of TARTES weighted with the energy flux for a fixed angle at SZA = 53 degrees. The point is that the albedo of TARTES increases with SZA, but that this effect is negated if the spectral albedos of TARTES are weighted with DISORT. The energy fluxes provided by DISORT shift more toward longer wavelengths for which the albedo is low. Therefore, if the spectral albedo calculated by TARTES is converted to a broadband albedo, the albedo does not increase as quickly as one would expect if you would use a fixed energy flux (compare the solid black line with the dashed line in Figure 11). We have changed our method from using DISORT to the pseudo-spherical SDISORT, for which the effect is less pronounced, but still visible. We have also tried to clarify this more clearly in the text: Page 19: This spectral shift is not or not sufficiently included in PKM and RACMO2 (Figure 11). For high SZA, DISORT models a clear spectral shift towards longer wavelengths, limiting the increase of the broadband albedo. If this effect is left out (black dashed line) the broadband albedo is much higher. Hence, the difference between the black solid and dashed line indicates this albedo decrease is not induced by the RW-approach, but by general red-shift in the incoming radiation. Page 20: The spectral albedo of TARTES is weighted with energy fluxes derived with DISORT (in black, solid line) or with the energy fluxes valid for a SZA of 53 degrees (black, dashed line) to compute a broadband albedo. Page 23: For clear-sky conditions during winter, i.e. large SZA, we show that the spectral shift towards larger wavelengths has substantial impact on the albedo, resulting in an albedo decrease.

Comment 4: Please, change 'assymetry' to 'asymmetry'

Done

Review #3

Comment 1a: DISORT is here run with 6 streams. Have you tested if that is enough to get the desired accuracy? If not, you should do so! Quote from Stamnes et al. (2000): "For strongly forward-peaked phase functions it is difficult to get accurate intensities with fewer than 16 streams, and even with 16 streams accuracy can be poor at some angles. Thus, careful users have been forced to use 32 or even 64 streams to be sure of getting 1% accuracy" Stamnes et al. here speak of intensities and not fluxes for which less streams are required, however, I expect that 6 streams are too little also to obtain very accurate fluxes. Tests should be done particularly for high solar zenith angles (SZAs), as these often occur in Greenland.

We thank the reviewer for this suggestion. After some tests, we conclude that the impact of the number of streams taken is very limited for this study. Nonetheless, we decided to rerun DISORT, so we now use 32 streams and some of the other suggestions that you have made. All results and figures now include DISORT run with 32 streams, even though the results and conclusions are hardly altered. Page 5: Thirty-two streams, i.e. computational polar angles, are used to solve the radiative transfer equation (Stamnes et al., 2000).

Comment 1b: In the DISORT simulations the surface broadband albedo is set to 0.5. In the supplementary scripts, it can be seen that this is done regardless of wavelength with the "albedo" input option in libRadtran. Here, the "albedo_file" input option should have been used, in which spectral albedos can be specified. In order to make the atmospheric and snowpack radiative transfer computations consistent, the TARTES spectral albedos should be used in this albedo file. As shown by for instance Nielsen et al. (GMD, 2014), the downward fluxes at the surface are not independent of the albedo. Given the complex variations of spectral irradiances and albedos shown in Fig. 1, it seems important to run DISORT with the TARTES albedos. Fig. 1 is a very illustrative figure by the way! An even better representation of the surface reflectance could be obtained by running coupled DISORT simulations for both the snowpack and the atmosphere. This can be done by adding the spectral inherent optical properties

of the layers of the snowpack as the lowest model layers in DISORT. In this way the full BRDF of the snow surface will be properly represented and coupled with the atmospheric simulations, which cannot be done with the two-stream TARTES simulations.

Although you are right that the chosen background albedo in DISORT is not very elegant, it turns out to have only a very limited impact on the representative wavelength and consequently on the narrowband albedo, as we have stated in section 2.3: "The surface broadband albedo of DISORT is set to 0.5, but as shown later, the sensitivity of both the surface broadband albedo and the aerosol load to the results is low", as well as in Figure 4. In Figure 4, we show that even if extreme values like 0 or 1 for the DISORT surface albedo are taken, that it matters insignificantly on the end result. One could introduce a more sophisticated surface albedo for DISORT as you propose, but in the end it will matter little, because it will certainly be in between the two extreme cases shown in Figure 4. Also, uncertainties in what surface albedo file to use if a more sophisticated profile is used.

Comment 1c: The "subartic winter" atmospheric profile is used. The reference describing the details of this is missing and should be added. I assume that this is one of the AFGL standard atmospheres of Anderson et al. (1986). Additionally, the "rural aerosols" of Shettle (1990) are used. How representative are these profiles for Greenland? Could typical atmospheric profile data from the CAMS reanalysis be used instead? The clear sky spectrum can change quite a lot depending on the gasses and aerosols assumed to be present. You should mention this uncertainty in the method chosen.

The uncertainties on the narrowband albedos of aerosols are small and are shown in Figure 4 for the extreme case of no aerosols, as well as for subarctic summer. More profiles have been tested, but omitted for clarity, because the weighted RMSE is similar. Using atmospheric data profile from the CAMS reanalysis for Greenland might be more typical, but will not result in any significant change in the results. The subarctic winter is indeed one of the AFGL standard atmospheres, and the reference is added

accordingly: Page 6: For the runs presented here, a subarctic winter atmospheric profile is chosen, which is one of the Air Force Geophysics Laboratory (AFGL) standards (Anderson et al., 1986).

Comment 1d: When inputting liquid and ice clouds to DISORT you assume these to have effective/equivalent radii of 10 $\mu$m and 20 $\mu$m, respectively. Here the former number is reasonable, but 20 $\mu$m is a very low number for typical ice clouds, where I would suggest using 50 $\mu$m instead. Also, you make these look-up table values a function of the cloud optical thickness rather than the cloud liquid water path (LWP) and ice water path (IWP). Since the cloud optical thickness is proportional to LWP+IWP and approximately inversely proportional to the effective/equivalent radii, similar relative changes in these cloud properties cause similar changes to the cloud optical thickness.

We do not agree with your statement that the effective radius of ice clouds that we have taken is too low for the Arctic and that we should use 50 $\mu$m instead. According to the following sources, the radius of ice clouds in the Arctic is typically between 10-30 (Stubenrauch et al., 2013; Fitzpatrick et al., 2004; Mahesh et al., 2001; Fu, 1996, Key et al., 2002, King et al., 2004). Therefore, we have taken 20 $\mu$m to work with. We have added these references in the paper: Page 6: ...which are realistic radii for clouds in the Arctic (Stubenrauch et al., 2013; Fitzpatrick et al., 2004; Mahesh et al., 2001; Fu, 1996, Key et al., 2002; King et al., 2004)

Regarding cloud optical thickness, see response of comment 2.

Comment 1e: In the DISORT experiments a range of cloud ice water path of up to 5 kg/m2 is simulated. This is at least 10x more than a realistic maximum value for clouds over Greenland. Cloud liquid water paths of up 40 kg/m2 are also simulated. This is also an order of magnitude higher that cloud water paths that can occur even in the tropics. I suggest that the simulations are done for more realistic ranges.

We are aware that the last elements for both IWP and LWP are very high, but we decided to do this for two reasons. Firstly, we also want RACMO2 to be able to work

with glaciers in lower latitudes, therefore we have extended the lookup tables to larger values than would be necessary for Greenland. Secondly, we want RACMO2 to always have some representative wavelength, even if it represents a non-physically high LWP or IWP. RACMO2 is therefore not expected to reach such high values for Greenland. Also keep in mind that this paper is to give an expression on how to couple spectral to narrowband albedos, and the lookup table shown in Figure 9 is only an example of how a final lookup table would look like. If our method is applied to another model, other values of IWP and LWP might be necessary. The following has been added in the paper accordingly: Page 17: ... of the most similar conditions. The lookup table of Figure 9 contains high values of IWP and LWP to allow RACMO2 to be run for lower latitudes and to ensure that RWs are always calculated, even if RACMO2 would produce unusually thick clouds.

Comment 1e continued: Also, the results shown in Fig. 7 are run for a LWP of 0.5 kg/m2. Is that a typical LWP for clouds over Greenland? Please update your experiments to more typical values

As you have requested, we have changed Figure 7 for LWP = 0.5 kg m2 to 0.1 kg m2, which occurs more often in Greenland and is consistent with Figure 6. However, the conclusions do not change. The following has also been changed in the paper: Page 13: In this figure, a LWP = 0.1 kg m2... Page 14, caption Figure 7: ...keeping LWP = 0.1 kg m2 constant.

Comment 1f: In the supplementary scripts it can be seen that the rte_solver disort is used in the libRadtran/DISORT experiments. Here, the rte_solver sdisort (Dahlback & Stamnes 1991) should be used instead. The regular disort/disort2 solver is designed for a plane-parallel geometry, where the atmosphere curves. sdisort is a pseudo-spherical disort solver, which accounts for the atmospheric curvature. In particular for high SZAs using disort will cause errors. This is also likely to explain the discrepancies seen for high SZAs in Fig. 11.

You are right and we have rerun DISORT, but now with SDISORT (and 32 streams). We have updated all figures accordingly. However, the results are still similar as before and the conclusions remain unchanged. Figure 11 is the only figure that changed somewhat. See Review #2, Comment 3 for the changes in the accompanying text. Other changes include: Page 5: ... at a given angle. The pseudo-spherical variant SDISORT (Dahlback and Stamnes, 1991) also accounts for atmospheric curvature, which is particularly relevant for high SZA. SDISORT is used in this paper and is called DISORT from now on unless stated otherwise. Page 6: SDISORT does not provide reliable fluxes for clouds with LWP or IWP > 1.0 kg m -2, hence the regular DISORT solver is used instead for these cases.

Comment 2: Page 21, lines 14-15: "However, this version of the IFS code embedded in RACMO2 does not explicitly model any cloud content properties nor includes a parameterization of the optical depth." That is incorrect! Since you have not included the RACMO2 source code in your supplementary material, I cannot tell what "the IFS code embedded in RACMO2" entails, but I am very familiar with the IFS radiation scheme and SRTM as used in cy33r1. In this the cloud optical thickness is parameterized. In fact it is computed for each spectral band in each 3D model grid box.

You are right that the cloud optical thickness is calculated in the IFS radiation scheme for each spectral band. We were not aware of this. Therefore, we decided to investigate the possibility to use the cloud optical thickness instead of LWP and IWP. However, after an extensive analysis, we concluded that the uncertainty that arises when using the cloud optical thickness is likely higher then if our LWP/IWP approach is applied. Our findings are described in a new section 3.5, and we have added the results in Figure 8.

In short, we tried the following methods to use the cloud optical thickness instead of LWP and IWP. Firstly, we tried to manually set the cloud optical thickness $\tau$ in DISORT to a realistic interval (King et al., 2004). Then we defined $\tau$ the same for every wavelength such that the cloud has a certain $\tau$. With this method, spectral effects are

neglected and therefore it is omitted. Secondly, we let DISORT calculate the optical thickness for the prescribed clouds used in the lookup tables. This would, in theory, allow us to reduce the lookup tables from LWP and IWP to $\tau$. However, it turns out that the type of cloud, i.e. liquid or ice, does have an effect on the spectral distribution computed by DISORT. In addition, the altitude of the cloud and the cloud effective radius have an impact on the spectral curve of DISORT and consequently on the calculated RWs (Note that the impact of cloud effective radius is also assessed in section 3.5). The extra figure (figure-3.png) illustrates how the representative wavelength RW alters considerably for band 8 for ice clouds, liquid clouds and everything in between.

Therefore, one would still require to make a distinction between ice and water clouds and has to define other cloud properties like the altitude. The difference between ice and liquid clouds is mostly caused by the various possible grain shapes and orientations of ice grains (King et al. 2004; Wyser and Yang, 1998). Also, a distinction has been made between ice and liquid water clouds in the IFS code, suggesting that is necessary to treat them differently.

However, it might still be possible to use cloud optical thickness in some form. Therefore, we have tried the following method. First, we derived RWs as function of the cloud optical thickness ($\tau$) as estimated by DISORT, using pure ice clouds and the IWPs as listed in Figure 9. Pure ice clouds were used as this type of clouds is most common in polar regions. Next, we derived $\tau$ with DISORT for all other cloud combinations in Figure 9 and linearly estimated the RWs for all these combinations using the RW-$\tau$ relations for pure ice clouds. Finally, we compared the subsequently derived narrowband albedos with the true narrowband albedos. Figure 8 shows that the weighted RMSE and bias of this $\tau$-approach is too high to be a viable option. Besides that, the quality of the liquid cloud optical thickness parameterization by Slingo (1989) in the IFS part of the ECMWF model version used in RACMO2 is limited and outdated (Hogan and Bozzo, 2018; Nielsen et al., 2014), and is updated in later iterations of the ECMWF model, but not available yet for RACMO2. Furthermore, the ice cloud parameterization by Fu 1996 (which is used in RACMO2) is not so reliable for thicker clouds above surfaces with a high albedo (Nielsen et al. 2014). To conclude, after a thorough investigation, using the cloud optical thickness as an alternative to LWP and IWP performs not as well as we have hoped for, so we decided to keep working with the method already described in the manuscript.

These considerations were incorporated in the manuscript in the following way: Page 15: 3.5 Cloud properties

LWP and IWP are chosen to represent the effect of clouds on the RW. In addition, microphysical properties of clouds such as the cloud effective radius r_e are known to impact the incoming radiation (Nielsen et al., 2014). We have chosen a realistic value of r_e, but in practice r_e will vary for each instance. Although the potential effect of r_e on the RWs is larger than BC and HULIS, it is still low (weighted RMSE < 0.01) for both clouds with small and large r_e, i.e. r_e,ice, r_e,liquid = 15, 5 and 30, 30 $\mu$m respectively (Figure 8). These values for r_e are on the lower and upper end of the probability range one could expect for the Arctic (King et al., 2004). Consequently, the typical weighted RMSE and bias is lower than indicated in Figure 8 and there is no need to make RWs dependend on r_e.

An alternative to the approach described in section 3.3 is the use of the cloud optical thickness $\tau$ instead of LWP and IWP to calculate RWs. This would be a valid approach if the spectral distribution is not altered considerably differently for ice clouds than for water clouds, as otherwise it would result in different RWs. Some differences between ice and liquid clouds are observed and are mostly caused by the various possible grain shapes and orientations of ice grains (King et al., 2004; Wyser and Yang, 1998). Still, a method using $\tau$ could be used if the uncertainty is small enough, but a choice regarding what type of clouds to calculate $\tau$ for, i.e. ice clouds, liquid water clouds or a combination, and its cloud properties has to be made nevertheless and will inevitably lead to uncertainties. We tested this "$\tau$ -approach", hence derived RWs as a function of $\tau$ for pure ice clouds, and linearly interpolated RWs for a given $\tau$ for liquid water clouds or

Interactive
comment

a combination of liquid water and ice clouds. The approach performs reasonably well (Figure 8), but the spread is large. If the statistical analysis of the "$\tau$ -approach" is limited to common LWPs and IWPs in the Arctic (< 1.0 kg m-2, see Figure 9), the RMSE is rather high (blue box and dark orange median in Figure 8), especially compared to the other parameters considered. Therefore, we decided not to use the cloud optical thickness as leading parameter to compute RWs. Page 14: Other factors controlling the narrowband albedo... Page 16: The blue box shows the 25th to 75th percentiles for cloudy conditions if limited to LWP and IWP < 1.0 kg m-2, with the dark orange line indicating the median. Low and high concentrations of BC and HULIS are considered, but the blue box is omitted for clarity. Page 16: The impact of cloud effective radius $r_e$ is evaluated for small and large values, i.e. $r_{e,ice}$, $r_{e,liquid}$ = 15, 5 and 30, 30 $\mu$m respectively. Finally, the $\tau$-approach is shown, as is described in section 3.5. Only cloudy conditions are considered for the cloud effective radius and $\tau$-approach. Page 22: Other properties can possibly affect the spectral albedo of snow, e.g. cloud top height, but their effect is deemed negligible compared to the other known uncertainties. Page 22: Using the cloud optical thickness instead of LWP and IWP to calculate RWs does not have the desired result, as a distinction between ice and liquid water clouds still have to be made. Moreover, the quality of the liquid cloud optical thickness parameterization by Slingo (1989) in the IFS part of the ECMWF model version used in RACMO2 is limited and outdated (Hogan and Bozzo, 2018; Nielsen et al., 2014), and is updated in later iterations of the ECMWF model, but not available yet for RACMO2. In addition, the ice cloud parameterization by Fu (1996), which is used in RACMO2, is not so reliable for thicker clouds above surfaces with a high albedo (Nielsen et al., 2014). Therefore, the use of cloud optical thickness in RACMO2 to determine RWs would result in an additional uncertainty on top of the described uncertainties in section 3.5. Hence, the option to use cloud optical thickness instead of LWP and IWP has been dismissed

Comment 3: Page 9, line 10: "The broadband albedo, which is for direct radiation close to 0.78 for most SZAs..." The broadband albedo of snow can be quite different

from 0.78 depending on atmospheric and snow conditions. Please correct this line to reflect this!

You are right that it is not clear what we meant. We have changed it accordingly: Page 9 The broadband albedo, which is for direct radiation and for the atmospheric and snow conditions described in the method section close to 0.78 for most SZAs (except for high SZA), and is used to compute a RMSE for each band.  

Minor comments: - Abstract, line 5: "... Integrated Forecast System atmospheric..." –> "... Integrated Forecast System (IFS) atmospheric..." Also, add the version number 33r1!

Done

- Abstract, line 10: "... 14 spectral bands of the ECMWF shortwave..." –> "... 14 spectral bands of the IFS shortwave..."

Done

- Page 1, lines 22-23: "... the melt-albedo feedback, e.g. Dumont et al. (2014)." –> "... the melt-albedo feedback (e.g. Dumont et al., 2014)."

...the melt-albedo feedback (e.g. Van As et al., 2013)."

- Page 2, line 4: "... solar angle" –> "... solar zenith angle"

Done

- Page 2: You need to add spectral band definitions of what you mean, when you refer to "near-UV" and "near-IR". ...is highest for near-ultraviolet (near-UV, 300-400 nm), visible and near-infrared (near-IR, 750-1400 nm) radiation...

- Page 2, lines 13-14: "... the ratio of upwards to downwards shortwave radiative flux integrated over the solar spectrum." –> "... the ratio of upwards to downwards shortwave radiative flux on a horizontal surface integrated over the solar spectrum."
Here you should also add explanations of the direct ("black sky") albedo, which varies as a function of the SZA, and the diffuse ("white sky") albedo. Both of these are used as input variables to SRTM in the IFS radiation scheme.

... the ratio of upwards to downwards shortwave radiative flux on a horizontal surface integrated over the solar spectrum.

Also, a distinction has to be made for the albedo of direct radiation, which varies as a function of the solar zenith angle (SZA), and of diffuse radiation. Although broadband albedo...

- Page 2, lines 19-20: "For example, a buried dark impurity layer will only significantly affect near-UV albedo." This I disagree with. Water has minimal absorption at the UV/violet spectral boundary, and the absorption is also very low in the UV, blue and green parts of the spectrum. Thus, these parts are also significantly affected by underlying impurities.

Done

- Page 2, line 21: "... the thermal regime" –> "... the snow heating rates"

Done

- Page 2, line 30: The RRTM_SW (also known as SRTM in the IFS code) references are placed after "RACMO2" in this line. They should be moved back to where RRTM_SW is referred to!

Done

- Page 4, lines 1-2: "RRTM_SW computes flux profiles for clear-sky and total-sky conditions on hourly intervals." –> "RRTM_SW computes instantaneous flux profiles for clear-sky and total-sky conditions."

Done

[Figure]

- Page 4, lines 14-15: "... specific surface area (SSA)..." This is the same acronym that is used for single-scattering albedo, an essential radiative transfer variable, which can be confusing. You should consider using something else.

In this manuscript, single-scattering albedo is not used, while specific surface area is an often occurring term. The abbreviation SSA is well defined and often used in other work, like Gallet et al. (2009) and Libois et al. (2014), so we decided to keep this acronym.

- Page 5: lines 31-32: "The effective droplet radius for ice and water clouds ..., which is a realistic radius for clouds" –> "The effective droplet radius for ice and water clouds ..., which are realistic radii for clouds"

Done

- Figure 12: This is a very nice figure, however, it is very difficult to distinguish cloud covers of 0 and 1. Please expand this part of the figure, so that these data are not hidden by the graph axes!

Done

Please also note the supplement to this comment:
https://www.geosci-model-dev-discuss.net/gmd-2018-175/gmd-2018-175-AC1-supplement.pdf

———————————————

[Figure]

[Figure]

**Fig. 1.** Updated Figure 11 from the manuscript

[Figure]

**Fig. 2.** Updated Figure 8 from the manuscript

[Figure]

**Fig. 3.** Extra figure which shows the RW as a function of cloud optical thickness for pure ice, pure liquid and both ice and liquid clouds

[Figure]

---

## Referee Report (RR1)

**The importance of the snow spectral albedo and integrated water vapour**

Kristian Pagh Nielsen, Danish Meteorological Institute

The revised version of the discussion paper: "A module to convert spectral to narrow-band snow albedo fro use in climate models: SNOWBAL v1.0" by C. T. van Dalum, W. J. van de Berg, Q. Libois, G. Picard and M. .R. van den Broeke will here be reviewed. I was anonymous reviewer #3 of the original submission, but will here not be anonymous since this review contains orginal material.

This review has been challenging to make since the authors on one hand have made a revised manuscript which addresses the issue of atmosphere surface coupling of snow spectral albedo at a level, which is higher than what has been included so far in other climate and weather models. On the other hand, there are still major issues with the manuscript, and the authors failed to follow many of the advices I gave in my first review. These are listed below with the same comment numbers as in the first review.

1b: It is an error that the authors have run their simulations with a spectrally constant albedo of 0.5. In my review I suggested to run with actual snow spectral albedos in stead - for instance from TARTES (Libois et al., 2013). The authors have not done so and replied:

> "Allthough you are right that the chosen background albedo in DISORT is not very elegant, it turns out to have only a very limited impact on the representative wavelength and consequently on the narrowband albedo, ..."

> "In Figure 4, we show that even if extreme values like 0 or 1 for the DISORT surface albedo are taken, that it matters insignificantly on the end results."

Here it appears that the authors have only tested running with spectrally constant albedos of 0 and 1. They fail to understand that it is the spectral variations of the albedo that are important for the representative wavelengths that they seek to determine. In Fig. 1 the importance of accounting for the snow albedo spectral variations is illustrated.

These spectra are computed with the libRadtran/DISORT package (Mayer & Kylling, 2005; Stamnes et al., 1988), which is also used by the authors. The 1

[Figure]

Figure 1: Surface net solar spectral irradiances computed for an altitude of 1.0 km, a solar zenith angle of $56°$, and an atmospheric water vapour load of 5 kg/m$^2$. The green curve shows the net irradiances for a constant surface albedo of 0.5. The red curve shows the net irradiances for the Jin et al. (2008) albedo spectrum of aggregate snow particles. The vertical lines show the wavelength band divisions between the spectral bands of the IFS 14 band shortwave radiation scheme.

nm resolution solar spectrum of Kurucz (1992) is used together with the psedo-spectral LOWTRAN/SBDART option (Pierluissi & Peng, 1985; Ricchiazzi et al., 1998). The ozone cross sections are given by Bass and Paur (1985). The default aerosols of Shettle (1989) are used with the AFGL subarctic summer atmosphere (Anderson et al., 1986). The atmospheric water vapour load is scaled to 5.0 kg/m$^2$.

In one of the simulations shown in Fig. 1 (the green curve) a constant spectral albedo of 0.5 is used. In the other simulation (the red curve) a typical snow albedo spectrum is used. Here a spectrum from Jin et al. (2008) for aggregated snow particles is used. Note that the net solar spectral irradiance at surface level is shown rather than the downward solar spectral irradiance, since it is the net irradiance that determines the snow melt. The 14 IFS shortwave (solar) spectral

bands are marked in Fig. 1 with black lines - or at least those of these spectral bands that are within the spectral range from 300 nm to 3000 nm. The major difference from using an actual spectral snow albedo, rather than just assuming the snow to be grey, on both the full spectrum and the band representative wavelengths is clear to see; using a constant albedo, as the authors have done is wrong. This needs to be fixed!

1c:   In my initial review I questioned using the AFGL "subarctic winter" standard atmosphere (Anderson et al., 1986). I pointed out that the clear sky spectrum changes quite a lot depending on the gases and aerosols present and suggested for instance using atmospheric profiles from the Copernicus Atmospheric Monitoring System (CAMS) dataset. The authors replied:

> "Using atmospheric data profile from the CAMS reanalysis for Greenland might be more typical, but will not result in any significant change in the results."

[Figure]

Figure 2: Surface net solar spectral irradiances computed for varying atmospheric water vapour loads. The Jin et al. (2008) albedo spectrum of aggregate snow particles is used. Otherwise, the computations and the figure have been made as Fig. 1.

That conclusion is just wrong and something must be wrong with the methodology of the authors when they find no significant change in their results for different atmospheric profiles. Here it suffices to test the solar spectral irradiance variations as a function of the atmospheric water vapour load, since water vapour is the primary gas affecting the solar irradiance. In Fig. 2 simulated surface net solar spectral irradiances are shown for atmospheric water vapour loads of 0.1 kg/m$^2$ (red curve), 1 kg/m$^2$ (green curve), 10 kg/m$^2$ (blue curve) and 40 kg/m$^2$ (magenta curve). The Jin et al. (2008) snow albedo spectrum is used. Again the IFS shortwave spectral band boundaries are marked on the figure with black lines. It is clear that if representative wavelengths are to be chosen for several of these bands, these must be different when the water vapour load changes.

[Figure]

Figure 3: As Fig. 2 but for a constant surface albedo of 0.5.

This is also the case if a constant spectral albedo of 0.5 is used as shown in Fig. 3, so something is wrong when the authors find no significant changes when they run their simulations with a different atmospheric profile.

In Fig. 4 the cumulative distribution function of hourly atmospheric water loads from the month of July 2012 for a region covering Greenland is shown. In the figure the water vapour loads of the 6 AFGL atmospheric profiles are also marked

[Figure]

Figure 4: The cumulative distribution function of hourly atmospheric water loads from the ERA5 reanalysis dataset (Hersbach & Dee, 2016) for the month July 2012 and the longitude-latitude region between 75°W to 10°W, and 59°N to 86°N. This region covers Greenland and the surrounding areas. In the figure the water vapour loads in the AFGL standard atmospheres are marked with blue dashed lines.

with blue lines. In order of increasing water vapour load these are: Subarctic winter, midlatitude winter, U.S. standard, subarctic summer, midlatitude summer and tropical. The figure illustrates how variable the atmospheric water vapour load is in the Arctic.

To sum up the above results show that in order to simulate the shortwave (solar) contribution to snow melt accurately, the spectral effects of varying the atmospheric water vapour load must be accounted for!

Here it should be noted that the effects of water vapour on the solar spectrum have been known for a long time (e.g. Abbot, 1911), but there appears to have been a lack of awareness on this issue for snow albedo modelling. Thus, it is also not accounted for in the study of Gardner and Sharp (2010).

1d: The authors disagree with my comment that the effective radius of ice clouds that

they assume of 20 $\mu$m is too low:

> "We do not agree with your statement that the effective radius of ice clouds that have taken is too low for the Arctic and that we should use 50 $\mu$m instead. According to the following sources, the radius of ice clouds in the Arctic is typically between 10–30..."

...and then they list several papers including Fu (1996). Just checking the generalized mean effective sizes ($D_{ge}$) of cloud ice particles given in Table 2 of Fu (1996) shows this to be wrong. In this cloud ice particles of up to $D_{ge} = 130$ $\mu$m are listed, which corresponds to an effective radius of 85 $\mu$m. Also, in the source code of cloud ice effective radius used in the IFS cy33r1 radiation scheme, as used in the RACMO simulations (`.../phys_ec/radlswr.F90` lines 432-483), larger cloud ice effective radii are defined depending on the namelist integer variable `NRADIP`. Thus, if `NRADIP = 0` the cloud ice effective radii are fixed at 40 $\mu$m; if `NRADIP = 1` the minimum cloud ice effective radius is set to 40 $\mu$m; if `NRADIP = 2` or `NRADIP = 3` the minimum cloud ice effective radius is set to 30 $\mu$m. Thus, unless the authors have modified the IFS cy33r1 radiation scheme, their claim here is inconsistent with their own source code!

These three major issues need to be addressed properly for the paper to be accepted. The other major and minor issues that I brought up in my first review have been addressed to a satisfactory level.

**References**

Abbot, C. G., The solar constant of radiation, Proc. Amer. Phil. Soc., 50 (199), 235–245, 1911.

Anderson, G. P., Clough, S. A., Kneizys, F. X., Chetwynd, J. H., and Shettle, E. P.: AFGL Atmospheric Constituent Profiles (0–120 km), Tech. Rep. AFGL-TR-86-0110, Air Force Geophysics Lab Hanscom AFB, MA, USA, 1986.

Bass, A. M. and Paur, R. J.: The Ultraviolet Cross-Sections of Ozone: I. The Measurements, in: Atmospheric Ozone, edited by Zerefos, C. S. and Ghazi, A., pp. 606–610, Springer, Netherlands, 1985.

Fu, Q.: An accurate parameterization of the solar radiative properties of cirrus clouds for climate models, J. Climate, 9, 2058–2082, 1996.

Gardner, A. S. and Sharp, M. J.: A review of snow and ice albedo and the development of a new physically based broadband albedo parameterization, J. Geophys. Res., 115, F01009, https://doi.org/10.1029/2009JF001444, 2010.

Hersbach, H., Dee, D.: ERA5 reanalysis in production, ECMWF Newsletter, 147, 7, 2016.

Jin, Z., Charlock, T. P., Yang, P., Xia, Y., and Miller, W.: Snow optical properties for different particle shapes with application to snow grain size retrieval and MODIS/CERES radiance comparison over Antarctica, Rem. Sens. Environ., 112, 3563–3581, https://doi.org/10.1016/j.rse.2008.04.011, 2008.

Libois, Q., Picard, G., France, J. .L., Arnuad, L., Dumont, M., Carmagnola, C. M., and King, M. D.: Influence of grain shape on light penetration in snow, Cryosphere, 7, 1803–1818, https://doi.org/10.5194/tc-7-1803-2013, 2013.

Kurucz, R. L.: Synthetic infrared spectra, in: Infrared Solar Physics, IAU Symp. 154, edited by Rabin, D. M. and Jefferies, J. T., Kluwer, Acad., Norwell, MA, USA, 1992.

Mayer, B. and Kylling, A.: Technical note: The libRadtran software package for radiative transfer calculations – description and examples of use, Atmos. Chem. Phys. Discuss., 5, 1319–1381, 2005.

Pierluissi, J. H. and Peng, G.-S.: New molecular transmission band models for LOW-TRAN, Opt. Eng., 24, 541–547, 1985.

Shettle, E. P.: Models of aerosols, clouds and precipitation for atmospheric propagation studies, in: Atmospheric propagation in the UV, visible, IR and mm-region and related system aspects, 454, AGARD Conference Proceedings, 1989.

Stamnes, K., Tsay, S.-C., Wiscombe, W., and Jayaweera, K.: Numerically stable algorithm for discrete-ordinate-method radiative transfer in multiple scattering and emitting layered media, Appl. Opt., 27, 2502–2509, 1988.

---

## Author Response (AR2)

[revised manuscript text omitted]

by C.T. van Dalum et al.

We would thank the reviewer Christian Nielsen for his constructive comments that have improved the accuracy of the calculations and the clarity of the paper. In black the comment, in orange the response, in blue the changes.

Note that we have changed the title of our manuscript to represent the newest version of SNOWBAL, which is version 1.2. We would also like to have the title adjusted accordingly in the final publication, if the manuscript is accepted.

**Comment 1b:**
It is an error that the authors have run their simulations with a spectrally constant albedo of 0.5. In my review I suggested to run with actual snow spectral albedos instead - for instance from TARTES (Libois et al., 2013). The authors have not done so and replied:
"Although you are right that the chosen background albedo in DISORT is not very elegant, it turns out to have only a very limited impact on the representative wavelength and consequently on the narrowband albedo, . . ."
" Figure 4, we show that even if extreme values like 0 or 1 for the DISORT surface albedo are taken, that it matters insignificantly on the end results."

Here it appears that the authors have only tested running with spectrally constant albedos of 0 and 1. They fail to understand that it is the spectral variations of the albedo that are important for the representative wavelengths that they seek to determine. In Fig. 1 the importance of accounting for the snow albedo spectral variations is illustrated.
These spectra are computed with the libRadtran/DISORT package (Mayer & Kylling, 2005; Stamnes et al., 1988), which is also used by the authors. The 1 nm resolution solar spectrum of Kurucz (1992) is used together with the psedo-spectral LOWTRAN/SBDART option (Pierluissi & Peng, 1985; Ricchiazzi et al., 1998). The ozone cross sections are given by Bass and Paur (1985). The default aerosols of Shettle (1989) are used with the AFGL subarctic summer atmosphere (Anderson et al., 1986). The atmospheric water vapour load is scaled to 5.0 kg/m2.
In one of the simulations shown in Fig. 1 (the green curve) a constant spectral albedo of 0.5 is used. In the other simulation (the red curve) a typical snow albedo spectrum is used. Here a spectrum from Jin et al. (2008) for aggregated snow particles is used. Note that the net solar spectral irradiance at surface level is shown rather than the downward solar spectral irradiance, since it is the net irradiance that determines the snow melt. The 14 IFS shortwave (solar) spectral bands are marked in Fig. 1 with black lines - or at least those of these spectral bands that are within the spectral range from 300 nm to 3000 nm. The major difference from using an actual spectral snow albedo, rather than just assuming the snow to be grey, on both the full spectrum and the band representative wavelengths is clear to see; using a constant albedo, as the authors have done is wrong. This needs to be fixed!
After careful examination of the issues you described, we agree that the albedo chosen in DISORT does matter for the incoming radiation and therefore on the RWs. However, you described the problem using the net spectral irradiance, but we only use the downward flux. The albedo, and consequently a net flux, will be determined by TARTES, which is after all the point of the paper. In other words, we only use the DISORT downward radiation as weights for TARTES. Consequently, the albedo chosen in DISORT has only a limited effect. More specifically, the albedo in DISORT has no impact on the direct radiation provided by DISORT, as per definition, this is the flux directly coming from the sun (See Figure 1a). However, we failed to realize that it does have an impact on the incoming diffuse radiation, as some radiation is reflected by the surface, which can be both direct and diffuse radiation, and then scattered back in the form of diffuse radiation. Therefore, in some cases, the

diffuse flux can differ quite significantly because of the surface albedo (See Figure 1b), and we therefore agree with you that a different albedo has to be taken for DISORT.

Fortunately, TARTES is able to provide a spectral albedo curve suitable for DISORT. The question is, do we have to use a different albedo curve for given conditions? For diffuse radiation, TARTES compute an albedo curve as if it is a direct curve of 53 degrees. Therefore, even though the DISORT irradiance changes considerably for given angle for diffuse radiation, a single albedo curve is sufficient for most of the downward diffuse radiation. However, a part of the downward diffuse radiation in DISORT is due to scattering of direct radiation on the snow surface and then scattered back in the atmosphere. The albedo curve for direct radiation does vary with solar zenith angle, but unfortunately, DISORT does not allow the implementation of a separate albedo curve for the direct and diffuse flux. In other words, a suboptimal albedo curve for direct or diffuse radiation has to be chosen regardless. Still, it only has a minor relevance compared to the switch from a broadband albedo curve (alb = 0.5 for example) to a spectral albedo curve.

For the downward direct radiation we have seen that the albedo curve does not matter (it only matters for its contribution to the diffuse downward radiation), but for consistency, the same curve has been used. In conclusion, the albedo curve for 53 degrees is used for all conditions (this is the curve described in Figure 1 of the manuscript, with the same snow conditions).

We are also aware that we are neglecting second order effects of variations of the firn properties on the narrowband albedos. In the end, the RW-approach is capable to include the major factors influencing the narrowband albedo of snow, but cannot include all minor processes. On the other hand, the resulting uncertainties and possible biases are small in comparison to uncertainties in grain size evolution and observations of (narrowband) albedo.

All figures have been updated. In addition, paragraph 3.2 and Figure 4 of the Manuscript have been changed considerably.

[Figure]

**Figure 1**: *Downward solar flux for an albedo in DISORT of 0.5 and with a TARTES albedo curve for clear-sky conditions (a, left) and for cloudy conditions with LWP = 0.1 and IWP = 0.5 kg/m² (b, right).*

Page 6:

The surface albedo of DISORT is prescribed with a spectral albedo profile of TARTES. The surface albedo does not, per definition, matter for the direct downward radiative flux. As diffuse radiation is approximated as a direct beam with a SZA of 53 degrees by TARTES, such an albedo curve should be suitable for all SZAs for the diffuse downward radiative flux. Still, some part of the diffuse flux is due to direct radiation that is scattered back by the atmosphere after reflecting at the surface first. For this part, a spectral albedo curve depending on SZA would be more suitable, but limitations in DISORT prevent this. Besides, variations in the spectral albedo curve due to SZA only have a second order effect on DISORT with respect to other variables. Therefore, a spectral albedo profile of TARTES using a SZA of 53 degrees for the reference snowpack is sufficient enough to use as the surface albedo of DISORT.

**Comment 1c:**

In my initial review I questioned using the AFGL "subarctic winter" standard atmosphere (Anderson et al., 1986). I pointed out that the clear sky spectrum changes quite a lot depending on the gases and aerosols present and suggested for instance using atmospheric profiles from the Copernicus Atmospheric Monitoring System (CAMS) dataset. The authors replied:

"Using atmospheric data profile from the CAMS reanalysis for Greenland might be more typical, but will not result in any significant change in the results."

That conclusion is just wrong and something must be wrong with the methodology of the authors when they find no significant change in their results for different atmospheric profiles. Here it suffices to test the solar spectral irradiance variations as a function of the atmospheric water vapour load, since water vapour is the primary gas affecting the solar irradiance. In Fig. 2 simulated surface net solar spectral irradiances are shown for atmospheric water vapour loads of 0.1 kg/m2 (red curve), 1 kg/m2 (green curve), 10 kg/m2 (blue curve) and 40 kg/m2 (magenta curve). The Jin et al. (2008) snow albedo spectrum is used. Again the IFS shortwave spectral band boundaries are marked on the figure with black

lines. It is clear that if representative wavelengths are to be chosen for several of these bands, these must be different when the water vapour load changes. This is also the case if a constant spectral albedo of 0.5 is used as shown in Fig. 3, so something is wrong when the authors find no significant changes when they run their simulations with a different atmospheric profile.

In Fig. 4 the cumulative distribution function of hourly atmospheric water loads from the month of July 2012 for a region covering Greenland is shown. In the figure the water vapour loads of the 6 AFGL atmospheric profiles are also with blue lines. In order of increasing water vapour load these are: Subarctic winter, midlatitude winter, U.S. standard, subarctic summer, midlatitude summer and tropical. The figure illustrates how variable the atmospheric water vapour load is in the Arctic.

To sum up the above results show that in order to simulate the shortwave (solar) contribution to snow melt accurately, the spectral effects of varying the atmospheric water vapour load must be accounted for!

Here it should be noted that the effects of water vapour on the solar spectrum have been known for a long time (e.g. Abbot, 1911), but there appears to have been a lack of awareness on this issue for snow albedo modelling. Thus, it is also not accounted for in the study of Gardner and Sharp (2010).

After testing some of the statements you have made, we have to conclude that you are right. We did not realize that the variability in water vapour is too important to be neglected. Nevertheless, we need to keep the size of the look-up tables manageable. Luckily, the impact of precipitable water (PW) is only really relevant for the near-IR part of the spectrum, resulting in that it only significantly impacts the direct radiation (Figure 4 in the manuscript, which is updated to a new figure). Although the narrowband albedo of band 8 is altered considerably, it is irrelevant for diffuse radiation, as most near-IR radiation is already filtered out. This holds for clear-sky conditions as well as thick clouds (Figure 2, left figure). For very thin clouds, a larger near-IR radiative flux is observed for diffuse radiation as more direct near-IR radiation is scattered, but not yet completely absorbed (Figure 2, right figure). The broadband albedo difference between a PW of 4 kg m$^{-2}$, which is approximately subarctic winter, and 13 kg m$^{-2}$, which is close to the upper limit for which very thin clouds still occur quite (Figure 3), is larger for thin clouds than for thicker clouds.

[Figure]

**Figure 2**: *Diffuse irradiance and albedo as a function of wavelength for cloudy conditions for various PW values, for a thick cloud LWP = 0.2 and IWP = 0.1 kg m⁻². (left), for a thin cloud with LWP = 0.05 and IWP = 0.01 kg m⁻². (right). The albedo is derived by TARTES for a fresh snow layer and the irradiance by DISORT. The narrowband albedo for each of the first twelve spectral bands of RACMO2 is indicated by the dashed line. The black vertical lines on the x-axis indicate the spectral band edges of RACMO2. The horizontal coloured lines on the y-axis indicate the weighted broadband albedo. The water vapour is distributed vertically in the same manner as the subarctic winter, which has a PW of approximately 4 kg m⁻².*

More specifically, the broadband albedo for 4, 13 and 40 kg m⁻² PW for the cloud with LWP = 0.2 and IWP = 0.1 kg m⁻² is [0.9510, 0.9555, 0.9590] respectively, for LWP = 0.05 and IWP = 0.01 kg m⁻² is [0.8978, 0.9054, 0.9124]. In other words, the difference between 4 and 13 kg m⁻² PW for the thick cloud is 0.0045 and for the thin cloud 0.007, which is still small enough to neglect. Especially compared to the albedo difference between 4 and 13 kg m⁻² PW for direct radiation, which is more than 0.01. In addition, the albedo difference for direct radiation between 2 and 4 kg m⁻² PW is already 0.008, while the difference between 2 and 4 kg m⁻² PW is negligible for cloudy conditions.

[Figure]

**Figure 3**: *Distribution of PW in RACMO2 for 2012 for a LWP range between 0.025 and 0.075, and an IWP range between 0.005 and 0.025. This range represents the thinnest clouds still represented in the lookup table produced by SNOWBAL (See Figure 9 of the manuscript).*

Therefore, the issue of water vapour becomes manageable, as we only have to extent the lookup table for direct radiation. The lookup table for direct radiation now consists two dimensions, the PW and SZA, and is therefore treated similarly to the lookup table for cloudy conditions (i.e. the lookup tables with LWP and IWP). The following PW values (in kg m⁻²) are considered: [0.5, 2.0, 4.0, 6.0, 8.0, 10.0, 13.0,1 8.0, 25.0, 40.0]. This increases the computational time and memory usage, as the lookup tables

are now much larger for direct radiation, and a 2-d interpolation has to be done now. In the end, however, the conclusions do not change, but still all figures have been updated.

Page 1:
…effect of clouds, water vapour, snow impurities…

Page 1:
…on the solar zenith angle (SZA), cloud content and water vapour.

[revised manuscript text omitted]

**Comment 1d:**

The authors disagree with my comment that the effective radius of ice clouds that they assume of 20 micrometer is too low:

"We do not agree with your statement that the effective radius of ice clouds that have taken is too low for the Arctic and that we should use 50 micrometer instead. According to the following sources, the radius of ice clouds in the Arctic is typically between 10-30 micrometer. . . "

. . . and then they list several papers including Fu (1996). Just checking the generalized mean effective sizes (**D**ge) of cloud ice particles given in Table 2 of Fu (1996) shows this to be wrong. In this cloud ice particles of up to **D**ge = 130 micrometer are listed, which corresponds to an effective radius of 85 micrometer. Also, in the source code of cloud ice effective radius used in the IFS cy33r1 radiation scheme, as used in the RACMO simulations (.../phys_ec/radlswr.F90 lines 432-483), larger cloud ice effective radii are defined depending on the namelist integer variable NRADIP. Thus, if NRADIP = 0 the cloud ice effective radii are fixed at 40 micrometer; if NRADIP = 1 the minimum cloud ice effective radius is set to 40 micrometer; if NRADIP = 2 or NRADIP = 3 the minimum cloud ice effective radius is set to 30 micrometer. Thus, unless the authors have modified the IFS cy33r1 radiation scheme, their claim here is inconsistent with their own source code!

These three major issues need to be addressed properly for the paper to be accepted. The other major and minor issues that I brought up in my first review have been addressed to a satisfactory level.

Unfortunately, we still do not agree with the statement that the effective radius for ice clouds of 20 micrometer is too low. Let us therefore explain it in more detail.

In the last two decades, quite a few publications came out regarding effective radius of ice clouds, discussing the occurrence of ice clouds with a broader range of effective radii than in the relatively old paper of Fu et al. (1996). In the work of Fu et al. (1996), a relatively large effective radius is described for cirrus clouds, but the important side note has to be made that many of their measuring instruments were not able to measure particles smaller than 20-40 micrometer. Since then, the measuring range and accuracy has improved considerably.

For example, King et al. (2004) describe a remote sensing measurement series of cloud optical thickness and effective radius for liquid and ice water clouds in the Arctic. In their Figure 13d (Figure 5), they show that range of occurrence of effective radius in ice clouds is between 5 and 30 micrometer, with a peak around 10 micrometers. This indicates that 85 micrometers, as you suggested to take, is way too high, even 20 micrometers is on the higher end.

[Figure]

**Figure 5:** *Marginal probability density function of cloud optical thickness and effective radius for all (a), (b) water and (c), (d) ice pixels in the MAS flight line on 4 Jun 1998. The pair of probability density functions in each panel corresponds to the probability distribution of cloud retrievals for the 0.87- and 2.13-mm algorithm (blue distribution) and the 1.62- and 2.13-mm algorithm (red distribution). The distributions arising from changing the surface albedo at 0.87 mm from 0.6 to 0.5 or 0.7 are shown as solid curves in all panels. From King et al. (2004), Figure 13d.*

Another extensive research about clouds in the Earth system is done by Stubenrauch et al. (2013). Using multiple data sets, they have described a normalized frequency distribution of ice cloud effective radius (CREI) in their Figure 4b (Figure 6). They show that the distribution of occurrence is between 5 and 50 micrometer approximately, with the average varying between 22 and 32 micrometers.

In addition, Fitzpatrick et al. (2003) stated: *"the effective radii of crystals in ice clouds are typically 10–30 micrometer (Mahesh et al. 2001; Fu 1996; C. Schmidt and A. Heymsfield 2002, personal communication)"*.

[Figure]

**Figure 6:** *Normalized frequency distributions of cloud properties of I and W: CRE, CWP, and COD. Their global averages are indicated below the distributions. Statistics are averaged over daytime measurements (1330–1500 LT, except ATSR-GRAPE at 1030 LT). From Stubenrauch et al. (2013), Figure 4b.*

Therefore, we conclude that our taken value of 20 micrometer for the ice cloud effective radius is a fair value to take. For more information, see also the other added references.
Regarding the IFS ice cloud effective radius, it is true that RACMO with its current settings of NRADIP = 3 is limited to a minimum of 30 micrometer. However, as this publication is not necessarily focused on implementation in RACMO, but more on the general methods, it does not mean that we should take wrong values for the ice cloud effective radius just because RACMO cannot handle them. Besides, the IFS cycle in RACMO will be updated in the near future.

---

## Author Response (AR3)

[revised manuscript text omitted]

by C.T. van Dalum et al.

We thank Christian Nielsen for his review. We have made all changes that you have requested.
In black the comment, in blue the changes.

**Comments**
- Page 6, lines 13-14: "For this part, a spectral albedo curve depending on SZA would be more suitable, but limitations in DISORT prevent this." DISORT does not have such limitations, in particular with the improvements made to DISORT3 (Lin et al. 2015), this radiative transfer solver can handle the full Bi-directional Reflectance Distributions Function (BRDF) representation of surface reflectance. The words about "limitations in DISORT" should be removed from the text!

Page 6:
For this part, a spectral albedo curve depending on SZA would be more suitable, but would have a second order effect on DISORT with respect to other variables.

-#-#-#

- Page 11, line 17: "In DISORT, the PW can be varied..." DISORT is a radiative transfer solver that takes layered optical properties as input. Integrated water vapour and other physical quantities is not varied in DISORT. The authors have used DISORT as fed by the libRadtran scripting system, which ensures that the physical quantities are computed into optical properties for the DISORT solver. Thus, it is in the libRadtran scripting system that the integrated water vapour can be varied. Here, and elsewhere, the text should be corrected to separate the preprocessing done in libRadtran and the computations in DISORT.

Page 5:
The Discrete Ordinate Radiative Transfer (DISORT) solver (Stamnes et al., 1988, 2000) computes a net shortwave radiation flux at the surface for direct and diffuse radiation for atmospheric conditions that are prescribed using the libRadtran software package (Mayer and Kylling, 2005).

Page 6:
Both the subarctic winter atmospheric profile and the aerosol type and load are the closest representation for the Arctic that we can describe in the libRadtran package. The surface albedo of DISORT is prescribed in libRadtran with a spectral albedo profile of TARTES.

Page 11:
Here, rural winter aerosols for a subarctic winter atmospheric profile are prescribed in the libRadtran package.

Page 11:
In libRadtran, the IWV can be varied while still retaining the relative vertical distribution of water vapour of the subarctic winter.

Page 21:
The surface pressure in libRadtran is set to 900 hPa

-#-#-#

- Section 3.2 and elsewhere: The term "Precipitable water/PW", should be replaced with: "Integrated water vapour/IWV". The authors already use the correct term "water vapour" elsewhere in the manuscript. The term "precipitable water" is physically incorrect, since not all water vapour in an atmospheric column is "precipitable". Also, when precipitation occurs it is often gathered from a large area surrounding each model column, e.g. via a convective cell or cyclone structure. It does not only come from the column itself, which the term "precipitable water" implies. The term "integrated water vapour" is an accurate description of this variable.

Page 11:
…and as such, the impact of the vertically integrated water vapour (IWV) on the narrowband albedo…

In addition, we have changed all occurrences of PW to IWV, which is on page 11, 12, 16, 17, 18, 19 and 24, and in Figure 4.